# A second photoactivatable state of the anion-conducting channelrhodopsin *Gt*ACR1 empowers persistent activity

Kristin Labudda [1,2], Mohamad Javad Norahan[1,2], Lisa-Marie Hübner[1,2], Philipp Althoff[1,2], Klaus Gerwert[1,2], Mathias Lübben[1,2], Till Rudack [3,4] ✉ & Carsten Kötting [1,2] ✉

Optogenetics is a method to regulate cells, tissues and organisms using light. It is applied to study neurons and to develop diagnostic and therapeutic tools for neuron-related diseases. The cation-conducting channelrhodopsin ChR2 triggers photoinduced depolarization of neuronal cells but generates lower ion currents due to the *syn*-pathway of its branched photocycle. In contrast, the homologous anion-conducting ACR1 from *Guillardia theta* (*Gt*ACR1), exhibits high photocurrents. Here, we investigate the mechanistic cause for the observed high photocurrents in *Gt*ACR1 using FTIR spectroscopy. Unexpectedly, we discovered that the O intermediate of *Gt*ACR1 is photoactivable, allowing for fast and efficient channel reopening. Our vibrational spectra show a photocyclic reaction sequence after O excitation similar to the ground state photocycle but with slightly altered channel conformation and protonation states. Our results provide deeper insights into the gating mechanism of channelrhodopsins and pave the way to advance the development of optimized optogenetic tools in future.

Microbial rhodopsins are integral membrane proteins with seven trans-membrane helices and the cofactor retinal. Due to the photoactivation of the retinal, they can fulfill several functions, including proton pumps, ion pumps, and ion channels. The exact atomic mechanisms within the photocycles have been of scientific interest for decades. Starting with the "hydrogen atom of biophysics" bacteriorhodopsin[1–11], the research has expanded to all kinds of microbial rhodopsins, such as halorhodopsin[12–14], schizorhodopsins[15–19], and channelrhodopsins[20–23].

The identification and characterization of natural channelrhodopsins, such as the cation-translocating channelrhodopsin 2 (*Cr*ChR2) found in the chlorophyte *Chlamydomonas reinhardtii*, marked a milestone for optogenetics[24–29], a research field that utilizes genetically modified eukaryotic cells to elicit physiological effects triggered by visible or ultraviolet light[30–33]. For example, ectopically expressed *Cr*ChR2 can control action potential firing with high temporal and spatial resolution in mammalian neurons[34,35]. In addition to neurons, a variety of other cell types, such as muscle cells[36,37], immune cells[38,39], endocrine cells[36,40], or stem cells[41,42] can also be controlled by optogenetic tools. Recent advances highlight the vast potential of optogenetics for various medical applications, including the treatment of blindness[30,43], Parkinson's disease[44,45], cardiac arrhythmia[46,47],

and the regulation of insulin secretion[36,40]. Understanding the molecular mechanisms underlying channelrhodopsins is crucial for improving these proteins as optogenetic tools through rational design. In parallel, researchers are seeking agents with naturally higher photochemical efficiency and improved properties. For instance, anion-conducting channelrhodopsins (ACRs), which belong to the same superfamily as cation-conducting channelrhodopsins—namely, microbial rhodopsins—but differ in their ion selectivity[48,49]. Among the anion-conducting channelrhodopsins, anion channelrhodopsin-1 (*Gt*ACR1) from the cryptophyte *Guillardia theta* is so far the best characterized ACR in terms of its gating mechanism and photochemical reaction cycle[50–52]. The use of heterologously expressed *Gt*ACRs[53,54] has facilitated the application of optogenetic tools for neuron suppression via anion-mediated cellular hyperpolarization[50,55,56].

Both members of the microbial rhodopsin superfamily, *Cr*ChR2[57] and *Gt*ACR1[55,58] share the common topology of seven transmembrane α-helices (TM1–TM7), with the chromophore retinal bound to a lysine side chain[59–62], and a high sequence identity in the central constriction site (see Fig. 1a), representing the functional core of the protein, suggesting that they operate by similar reaction mechanisms. *Cr*ChR2 has been extensively studied using electrophysiological and spectroscopic methods[25,26,63–65]. In our previous

[1]Center for Protein Diagnostics (PRODI), Biospectroscopy, Ruhr University Bochum, Bochum, Germany. [2]Department of Biophysics, Ruhr University Bochum, Bochum, Germany. [3]Structural Bioinformatics Group, Regensburg Center for Biochemistry, Regensburg Center for Ultrafast Nanoscopy, University of Regensburg, Regensburg, Germany. [4]Biomolecular Simulations and Theoretical Biophysics Group, Faculty of Biology and Biotechnology, Ruhr University Bochum, Bochum, Germany. ✉e-mail: till.rudack@ur.de; carsten.koetting@rub.de

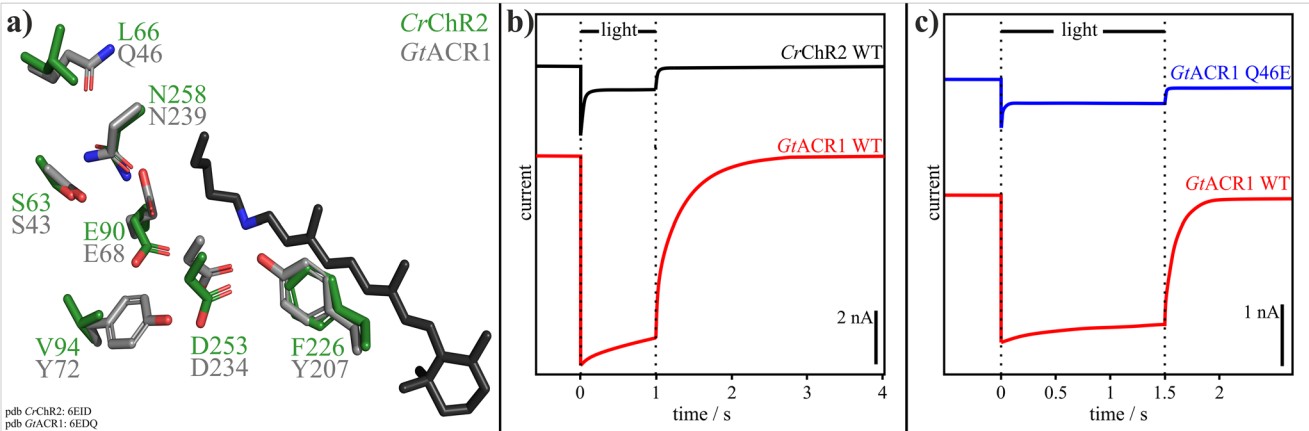

**Fig. 1 | Structural and functional comparison of *Cr*ChR2 and *Gt*ACR1.**
**a** Structural alignment of the central constriction site residues of *Cr*ChR2 (green amino acid carbon atoms) from PDB-ID 6EID[21] and *Gt*ACR1 (gray amino acid carbon atoms) from PDB-ID 6EDQ[22]. The retinal (black carbon atoms) is taken from *Gt*ACR1. Nitrogen atoms are colored blue and oxygen atoms red. **b** Photocurrents of *Cr*ChR2 WT and *Gt*ACR1 WT. The shown photocurrents are a response of *Cr*ChR2 and *Gt*ACR1 (data replotted from Govorunova et al.[48], Fig. 1D12) to a 1 s light pulse at a holding potential of −60 mV in HEK293FT cells 10–19 days after transfection. *Cr*ChR2 shows the characteristic decay of the photocurrent upon continuous illumination. *Gt*ACR1 produces a significantly larger photocurrent compared to *Cr*ChR2 and does not display the described inactivation of the photocurrent. However, channel closing is much slower in *Gt*ACR1 compared to *Cr*ChR2. **c** Photocurrents of *Gt*ACR1 WT and *Gt*ACR1 Q46E (data replotted from Kim et al.[55], Extended Data Fig. 8). The shown photocurrents are a response to a 1.5 s light pulse in patch-clamp recordings of HEK293 cells 24–48 h after transfection. The *Gt*ACR1 WT shows a significantly larger photocurrent compared to the *Gt*ACR1 variant Q46E. Furthermore, *Gt*ACR1 Q46E exhibits strong current attenuation.

work, we investigated the physiological and biophysical mechanisms of *Cr*ChR2 with time-resolved Fourier transform infrared (FTIR) spectroscopy, a powerful technique for studying molecular processes and obtaining dynamic information at high spatial and temporal resolution[7,66–70]. For microbial rhodopsins, light absorption by retinal triggers a photocycle involving several spectroscopically distinguishable intermediates[71]. FTIR has been particularly effective in characterizing structural changes in the photointermediates[72]. In the case of *Cr*ChR2, blue light excitation ($\lambda = 470$ nm) induces two parallel photoreaction cycles (see Supplementary Fig. 1)[73]. One of these is the so-called "dark-adapted" *anti*-cycle, characterized by the exclusive occurrence of a C=N-*anti* configuration of retinal and a well-conducting open state that decays relatively quickly.

The other one is the slowly decaying "light-adapted" *syn*-cycle, characterized by the 13-*cis*, C=N-*syn* configuration of retinal and the presence of poorly conducting photoproducts[73]. Electrophysiological measurements show that under continuous illumination, the highly conductive *anti*-cycle is associated with a current peak that drops rapidly—referred to as attenuation—when *Cr*ChR2 switches to the low-conductivity *syn*-cycle[66,73] (see Fig. 1b). Both cycles are spectroscopically distinguishable due to their cycle-specific retinal configurations. The spectroscopic marker band of the *anti*-cycle is found at 1188 cm⁻¹, representing the C=N-*anti* retinal configuration, while the marker band for the *syn*-cycle is at 1154 cm⁻¹, corresponding to the C=N-*syn* retinal configuration[73].

In contrast to *Cr*ChR2, only a highly conductive photocycle actively operates in *Gt*ACR1[52,66]. The photocycle model by Sineshchekov et al.[52], based on electrophysiological data, was significantly improved by Dreier et al.[66] using FTIR-spectroscopic measurements (see Fig. 2). After photoexcitation ($\lambda = 480$–500 nm) of the ground state, retinal in *Gt*ACR1 isomerizes from its all-*trans* configuration to the 13-*cis* C=N-*anti* configuration, initiating the K intermediate. Within 450 ns, K decays to the non-conducting L₁/L₁' intermediate. As L₁/L₁' decays, channel opening occurs in a two-step process with fast (18 µs) and slow (1.9 ms) phases, leading to the conducting L₂ intermediate. Channel closing also occurs in two steps, with the mechanistic closing happening in the transition from L₂ to M (35 ms). The photocurrent shuts down completely upon formation of the N/O intermediate, following the M intermediate, with a half-life of 107 ms. Finally, *Gt*ACR1 relaxes from the N/O intermediate—a fast

equilibrium that is difficult to dissect spectroscopically—back to the ground state with a lifetime of 4.4 s[66].

There is no spectroscopic evidence for a *syn*-cycle, as its typical marker band at 1154 cm⁻¹ is absent in the *Gt*ACR1 WT[66]. This is also reflected in the electrophysiological measurements, where photocurrent amplitudes are much higher than those observed in *Cr*ChR2, without significant current attenuation[48,51,52]. High photocurrents and low current inactivation levels in channelrhodopsins are highly desirable for optogenetic applications, as both features enable efficient, targeted stimulation of neurons. A fundamental prerequisite for the rational design of optimized optogenetic tools is a detailed functional understanding of the molecular mechanisms underlying channel opening and closing at the atomic level.

Therefore, we investigated the mechanistic role of the central gate, which is crucial for ion conductivity in channelrhodopsins[74], by comparing the relatively ineffective *Cr*ChR2 to the highly effective *Gt*ACR1. To gain insights into the limiting characteristics of *Cr*ChR2, we performed mutagenesis on *Gt*ACR1 to convert it to a more *Cr*ChR2-like variant. In doing so, we unexpectedly discovered a second photoactivatable state in addition to the *Gt*ACR1 ground state, which allows fast and efficient channel reopening. This second photoactivatable state explains the sustained high conductivity observed in electrophysiological studies under continuous illumination.

## Results and discussion
### *Gt*ACR1 variant Q46E
To gain insights into the functional differences between *Cr*ChR2 and *Gt*ACR1, we compared the existing X-ray structures[57,58] of the central constriction sites, which represent the functional cores of both proteins, as shown in Fig. 1a. *Cr*ChR2 and *Gt*ACR1 exhibit high sequence identity in the central gate, with residues such as S63, E90, D253 and N258 in *Cr*ChR2 being conservatively substituted by S43, E68, D234 and N239 in *Gt*ACR1. Especially E90 in *Cr*ChR2 and E68 in *Gt*ACR1 are critical for the function of both proteins. E90 in *Cr*ChR2 is one of the key determinants of ion selectivity, and its deprotonation is linked to ion conductance after light adaptation[57,73,75–79]. In *Gt*ACR1, early deprotonation of E68 has been shown to be crucial for gate formation[51,52,80]. However, there are notable differences within the central constriction sites of *Cr*ChR2 and *Gt*ACR1. For example, L66, V94, and F226 in *Cr*ChR2 are replaced by Q46, Y72, and Y207 in

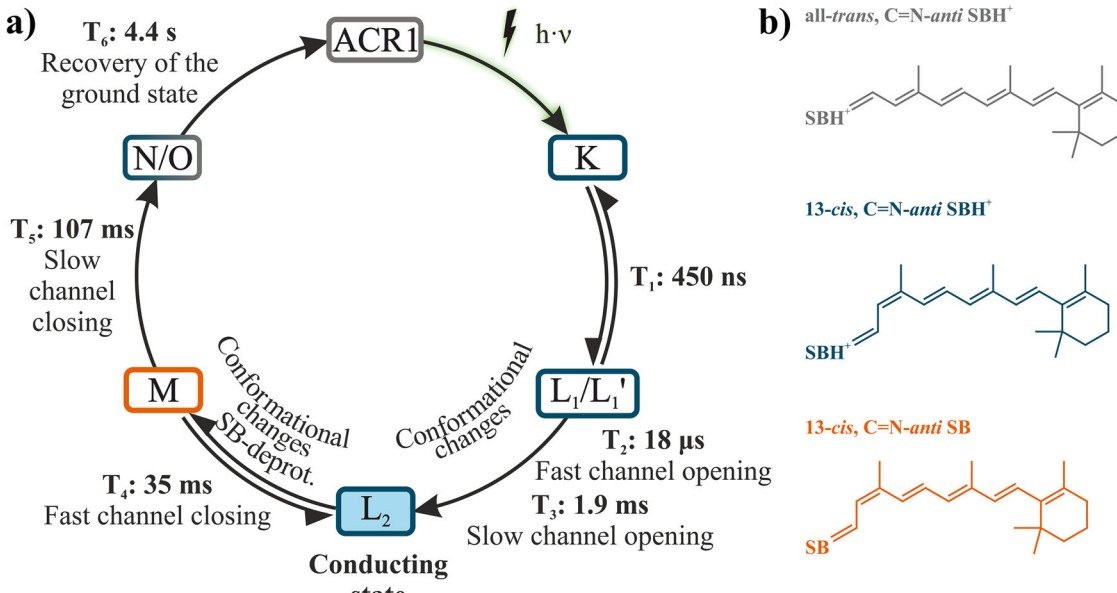

**Fig. 2 | Photocycle model of *Gt*ACR1. a** The photocycle model is modified after Fig. 4 from Dreier et al.[66]. After excitation of the *Gt*ACR1 ground state, the retinal isomerizes from all-*trans* to the 13-*cis* configuration, leading to the K Intermediate, which then decays to the non-conducting $L_1/L_1'$ intermediates (450 ns). Channel opening occurs in a two-step process with fast (18 µs) and slow (1.9 ms) channel opening, leading to the conducting $L_2$ intermediate (highlighted in blue). Channel closing takes place in the transition from $L_2$ to M (35 ms). Due to the equilibrium of $L_2$ and M, the photocurrent does not disappear completely until the formation of the N/O intermediate (107 ms). Finally, *Gt*ACR1 relaxes into the ground state from the N/O intermediate. **b** Illustration of the structural formulae of the retinal configurations that occur during the photocycle. Shown are all-*trans* C=N-*anti* retinal with protonated Schiff base (gray), 13-*cis* C=N-*anti* retinal with protonated Schiff base (blue), and 13-*cis* C=N-*anti* retinal with deprotonated Schiff base (orange). The colors of the boxes in A match the colors of the corresponding retinal conformation.

*Gt*ACR1, making these latter residues promising targets for investigating the limiting characteristics of *Cr*ChR2.

Among the residues of the central constriction site that are potential candidates responsible for attenuation, we focused on the most promising *Gt*ACR1 variant, Q46E, which exhibits electrophysiological features comparable to the wild type of *Cr*ChR2[48] (see Fig. 1c). Under continuous illumination, the variant displays a sharp current peak followed by current attenuation, which—similar to *Cr*ChR2[48]—might be due to the occurrence of a low conducting *syn*-cycle in this variant. To test this hypothesis, rapid-scan FTIR measurements were carried out at ambient temperature, with a neutral pH (7.5), and compared to wild-type measurements.

The major difference between the WT and the variant lies in the reaction rates, expressed in the half-lives derived from global fit analysis[81–84], as the *Gt*ACR1 variant Q46E is significantly slower than the wild type. In this case, only the half-lives $T_4$–$T_6$ were spectroscopically distinguished, as rapid-scan measurements were performed and $T_1$–$T_3$ fall outside the time-resolution limit. As summarized in Supplementary Table 1, the variant's half-life $T_4$ is about five times slower than the WT, $T_5$ is about seven times slower, and $T_6$ is about twice as slow as the WT.

Interestingly, the course of the key marker bands identified so far (see Supplementary Table 2 for a list of marker bands) of *Gt*ACR1 WT and its variant Q46E remains similar (see Fig. 3a, b). The conformational changes during channel opening and closing in the amide I region, visible at 1644 cm$^{-1}$, as well as channel opening itself and the conducting state, visible at 1691 cm$^{-1}$, follow the same pattern as described in literature for the wild type[66]. Furthermore, the protonated 13-*cis* retinal (1184 cm$^{-1}$) and the retinal C=C vibrations in the ground state (1529 cm$^{-1}$) could be readily followed.

Overall, the entire mid-IR amplitude spectra from 1000-1800 cm$^{-1}$ for *Gt*ACR1 WT and *Gt*ACR1 Q46E are almost identical (see Fig. 3c), indicating nearly indistinguishable photocycle processes for both proteins. The only notable difference is a shift in the marker band of the protonated Schiff base in 13-*cis* configuration, which is shifted from 1184 to 1180 cm$^{-1}$ in the $T_6$ FTIR amplitude spectra. This shift could be explained by the altered electrostatic environment resulting from the change from glutamine to glutamate in the central constriction site.

Most importantly, in both the WT and the Q46E variant of *Gt*ACR1, the band at 1154 cm$^{-1}$, which is typical for a 13-*cis* C=N-*syn*-configuration in *Cr*ChR2, is absent (see Fig. 3c). This clearly contradicts our initial assumption that a *syn*-cycle is responsible for the attenuated current in Q46E. The observed reduction in photocurrents in the Q46E variant must therefore be due to another cause. The extensive similarity between the UV/Vis and FTIR spectra suggests that *Gt*ACR1 WT and Q46E exhibit similar reaction mechanisms, except for the significantly decelerated photocyclic reaction rates in the variant. Since a *syn*-cycle is ruled out, and the conductive state of the channel is relatively short (even the slow channel-closing lifetime is about 290 ms), compared to the long return time to the excitable ground state, due to the increased $T_4$–$T_6$ half-lives, we conclude that the electrophysiologically observed attenuation is due to the accumulation of intermediates preceding the photoactivatable ground state.

Given the above explanation, it is questionable why attenuation is almost absent in *Gt*ACR1 WT upon continuous illumination (see Fig. 3). The channel closes completely with a half-life of only 107 ms, compared to the long half-life of $T_6$ = 4.4 s for the return to the ground state. This should immediately enrich the non-conducting N/O intermediate, leading inevitably to conductive attenuation. However, as no attenuation occurs in WT *Gt*ACR1, we hypothesize that continuous photocycling is driven by the N/O intermediate, specifically through photoactivation of the O intermediate, whose retinal cofactor is already in the all-*trans* configuration. This is a precondition that is otherwise only satisfied by the dark ground state.

## Repeated photoexcitability of the relaxing state

To validate our new theory on the photocycle model and the potential initiation of another photocycle from the N/O intermediate—more specifically, from the O intermediate—we conducted additional investigations on the *Gt*ACR1 WT. These experiments provided more insights into the gating mechanism of *Gt*ACR1 WT, as detailed below.

Under our measurement conditions, all molecules are excited, as demonstrated by experiments at lower laser intensities, resulting in almost

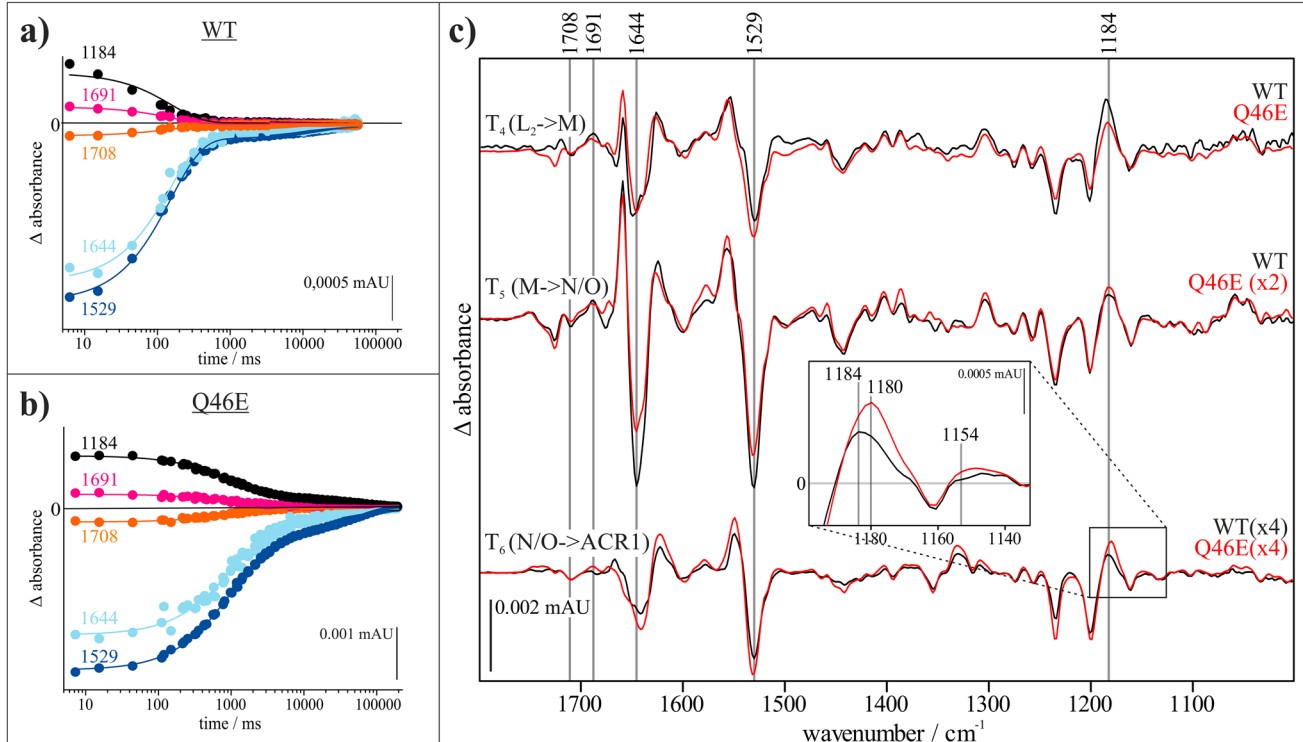

**Fig. 3 | Comparison of the spectroscopic data from *Gt*ACR1 WT and its variant Q46E. a, b** Time course of *Gt*ACR1 marker absorption bands in the *Gt*ACR1 WT and the variant Q46E. The time-resolved data were obtained from rapid-scan FTIR difference spectroscopic measurements. The band assignment is based on the work by Dreier et al.[66] assigning the 1644 cm$^{-1}$ band (light blue) to conformational changes during channel opening and closing in the amide I region, the 1691 cm$^{-1}$ band (pink) to channel opening and therefore the conducting state, the 1184 cm$^{-1}$ band (black) to protonated 13-*cis* retinal and the 1529 cm$^{-1}$ band (orange) to the retinal C=C vibrations in the ground state. **c** Comparison of the amplitude spectra from *Gt*ACR1 WT (black) and *Gt*ACR1 Q46E (red). Shown are the reactions from L$_2$ to M (WT $T_4$ = 43.7 ms, Q46E $T_4$ = 291.74 ms), from M to N/O (WT $T_5$ = 149.95 ms, Q46E $T_5$ = 1145.42 ms) and from N/O to ground state (WT $T_6$ = 15.9 s, Q46E $T_6$ = 32.66 s). Due to the low amplitude of some spectra, they have been amplified as indicated. An enlarged extract of the *Gt*ACR1 amplitude spectra at $T_6$ of the reaction of *Gt*ACR1 WT and *Gt*ACR1 Q46E is displayed, showing no absorption of a *syn*-marker band at 1154 cm$^{-1}$ in either the WT or the variant, and the shift from 1184 cm$^{-1}$ towards 1180 cm$^{-1}$ in the variant.

the same absorbance changes. Thus, we are working under saturation conditions. However, it must be taken into consideration that back-isomerization from the K intermediate (13-*cis*) to the initial all-*trans* state is possible[85] and a photo-equilibrium between these two states is established. We determined the position of the equilibrium by comparing the overall protein amount in the sample by means of the amide II absorption and the amount of protein undergoing the photocycle by means of the difference absorption of the retinal. This shows that about 41% of the sample enters the photocycle (see Supplementary Note 1). Upon a second excitation, the spectra would therefore contain a mixture of proteins excited from the O intermediate state and the ground state.

To test whether the hypothesis of additional excitability of the O intermediate applies to the *Gt*ACR1 WT, the proper timing for repeated excitation must be adjusted to ensure that the O intermediate is precisely targeted. Based on the known photocycle time constants, it was calculated that 1 s after the initial activation from the ground state, the M intermediate (which decays with a slow channel-closing half-life $T_5$ of 107 ms) is reduced to less than 1% of its initial value (see Supplementary Note 2), indicating that over 99% of the protein reached the N/O intermediate. Similarly, it was calculated that after 1 s of activation 96% of the proteins within the photocycle are still remaining in the N/O intermediate (see Supplementary Note 3).

At this time point, a second flash was applied to initiate another photocycle, which was monitored by time-resolved FTIR spectroscopy.

Under consideration that 41% of the proteins enter the photocycle, after 1 s, about 38% are in the N/O intermediate. Global fit analysis of the FTIR data obtained after the second laser flash revealed three distinct

reaction steps. The amplitude spectra for these steps are similar to the last three partial reactions ($T_4$ (L$_2$→M), $T_5$ (M→N/O) and $T_6$ (N/O→ACR1)) of the normal photocycle (see Fig. 4a)[66]. The half-lives of the reactions after the first and second flashes are comparable for $T_4$, but significantly slower for $T_5$ and $T_6$ following the second flash (see Fig. 4b, c and Supplementary Table 1).

The amplitude spectra of the partial reactions after the first and second laser flashes show largely identical absorption bands (see Fig. 4a), suggesting that the secondary triggered photocycle is very similar to the first and that rapid channel opening occurs. The most prominent differences between the spectra from single and repeated exposure are observed in the amide I and amide II regions (see Fig. 4a). Specifically, in $T_5$, shifts are observed from 1645 cm$^{-1}$ to 1635 cm$^{-1}$ and from 1659 cm$^{-1}$ to 1667 cm$^{-1}$, indicating conformational changes in the protein backbone, and from 1555 cm$^{-1}$ to 1562 cm$^{-1}$, indicating changes in the surrounding of the retinal. The most striking difference in the $T_5$ amplitude spectra is the band of protonated E68 at 1708 cm$^{-1}$. Although still visible, this band is largely reduced after the second flash. For $T_6$, shifts from 1640 cm$^{-1}$ to 1652 cm$^{-1}$ and from 1184 cm$^{-1}$ to 1180 cm$^{-1}$ occur (see Fig. 4a, f). The changes around 1640 cm$^{-1}$ are also associated with conformational changes in the protein backbone, while the shift of the 1184 cm$^{-1}$ band suggests alterations in the environment of protonated retinal in the 13-*cis* C=N-*anti*-SBH$^+$ configuration. The 1691 cm$^{-1}$ band, which is attributed to conformational changes during pore formation, shows an increase in absorption with repeated excitation (see Fig. 4d).

As calculated above, after 1 s, about 2/3 of the sample is in the ground state and 1/3 in the O intermediate. Therefore, the spectra contain a mixture

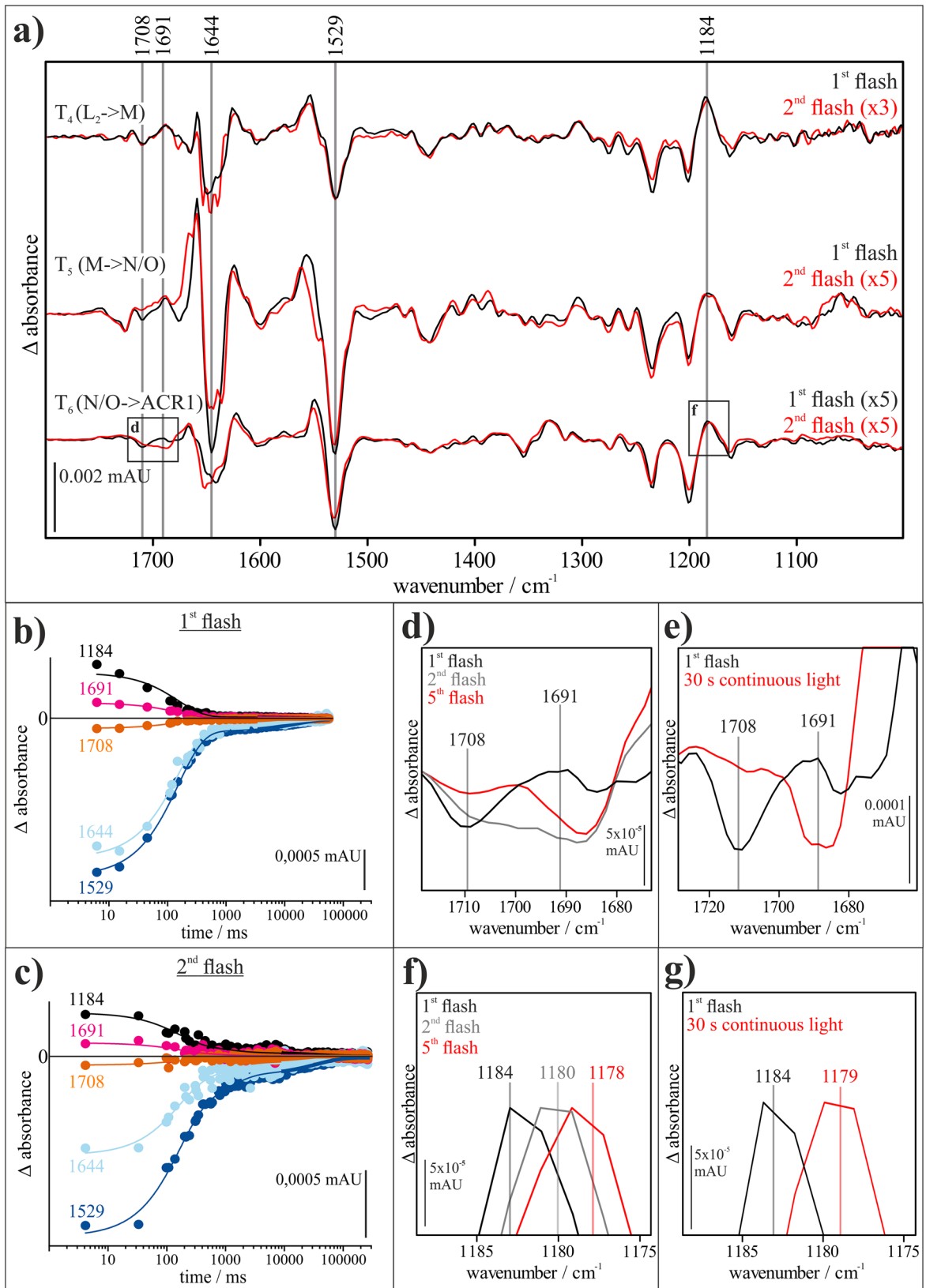

of proteins that were excited from the O intermediate and the ground state. To increase the amount in the O intermediate, spectral development was monitored under different illumination regimes: after one flash, two flashes, and five flashes (with 1 Hz repetition rate; see Fig. 4d, f), as well as after 30 s of continuous illumination (see Fig. 4e, g). With the 5th laser flash, about

87% of the sample is excited from the O intermediate (Supplementary Note 1).

Illumination after two and five flashes, as well as after 30 s, resulted in comparable half-lives of the reactions, although $T_4$ and $T_5$ were slightly slower under continuous illumination (see Supplementary Table 1). In the

**Fig. 4 | Comparison of single-flash and multi-flash experiments of GtACR1 WT.** **a** GtACR1 amplitude spectra of the reaction after the first flash (black) and the second flash (red). Shown are the reactions from $L_2$ to M (1st flash $T_4$ = 44 ms, 2nd flash $T_4$ = 88 ms), from M to N/O (1st flash $T_5$ = 150 ms, 2nd flash $T_5$ = 1438 ms) and from N/O to ground state (1st flash $T_6$ = 16 s, 2nd flash $T_6$ = 93 s). Due to the low amplitude of the $T_6$ spectra and the spectra after the second flash, they have been amplified as indicated. **b**, **c** Time course of GtACR1 marker absorption bands after one flash (**b**) and after the second flash (**c**). The time-resolved data were obtained from rapid-scan FTIR difference spectroscopic measurements. The band assignment is based on the work by Dreier et al.[66] assigning the 1644 cm$^{-1}$ band (light blue) to conformational changes during channel opening and closing in the amide I region, the 1691 cm$^{-1}$ band (magenta) to channel opening and therefore the conducting state, the 1184 cm$^{-1}$ band (black) to protonated 13-cis retinal and the 1529 cm$^{-1}$ band (blue) to the retinal C=C vibrations in the ground state. **d** GtACR1 amplitude spectra at $T_6$ of the reaction after one flash (black, $t_{1/2}$ = 16 s), two

flashes (gray, $t_{1/2}$ = 93 s), and five flashes (red, $t_{1/2}$ = 69 s). Spectra were scaled to the 1531 cm$^{-1}$ marker band for all-trans retinal. A reduction in the amplitude of the band at 1708 cm$^{-1}$ and an amplification in the amplitude at 1691 cm$^{-1}$ can be observed. **e** GtACR1 amplitude spectra at $T_6$ of the reaction after one flash (black, $t_{1/2}$ = 16 s) and 30 s of continuous illumination (red, $t_{1/2}$ = 68 s). Spectra were scaled to the 1531 cm$^{-1}$ marker band for all-trans retinal. A reduction in the amplitude of the bands at 1708 cm$^{-1}$ and 1691 cm$^{-1}$ can be observed. **f** GtACR1 amplitude spectra at $T_6$ of the reaction after one flash (black, $t_{1/2}$ = 16 s), two flashes (gray, $t_{1/2}$ = 93 s), and five flashes (red, $t_{1/2}$ = 69 s). Spectra were scaled to the 1184 cm$^{-1}$ marker band for protonated 13-cis retinal. A shift towards 1176 cm$^{-1}$ can be observed with a growing number of flashes. **g** GtACR1 amplitude spectra at $T_6$ of the reaction after one flash (black, $t_{1/2}$ = 16 s) and 30 s of continuous illumination (red, $t_{1/2}$ = 68 s). Spectra were scaled to the 1184 cm$^{-1}$ marker band for protonated 13-cis retinal. A shift towards 1179 cm$^{-1}$ can be observed.

amplitude spectra, the most prominent shift was observed at 1184 cm$^{-1}$ in the $T_6$ amplitude spectrum (see Fig. 4f, g), which shifted down to 1178 cm$^{-1}$ after the excitation with five flashes, while continuous illumination produced a shift to 1179 cm$^{-1}$. Furthermore, it was observed that the amplitude assigned to the E68 band at 1708 cm$^{-1}$ decreased further with increasing exposure time, while the band at 1691 cm$^{-1}$ became more pronounced. (see Fig. 4d, e).

In summary, the effects of repeated flash excitation or continuous illumination indicate various changes within the functionally active region, specifically the central gate of GtACR1.

The decrease in the E68 band at 1708 cm$^{-1}$ (see Fig. 4d, e) during the transition from the N/O to the ground state demonstrates a lower degree of reprotonation of E68 in the N/O intermediate with increased light exposure. Since the fast deprotonation of E68 is associated with the channel opening process[66,86], the deprotonated state of glutamate in the N/O intermediate may facilitate opening more readily than the state reached after full deprotonation and reprotonation, as observed after a single-flash activation. We assume that the central constriction site may therefore exist in a "pre-open state", which allows for rapid channel opening upon excitation from the O intermediate. Although the pre-open state does not conduct ions, there are structural features that are between the open state and the closed state, enabling the proteins to return faster to the conducting state.

This interpretation is further supported by the increased formation of the band at 1691 cm$^{-1}$ during stronger light exposure (see Fig. 4d, e). A higher absorbance at 1691 cm$^{-1}$ indicates an increased presence of the assigned group, in this case alterations in the secondary structure which only occur during channel opening, thereby supporting the buildup of the postulated pre-open state of the ion channel. The environmental changes may also affect the neighboring protonated Schiff base, which explains the observed shift of the band of protonated 13-cis C=N-anti retinal at 1184 cm$^{-1}$ (see Fig. 4f, g).

### Channel state after photoexcitation of the O intermediate

A major question is whether the state of the ion channel is altered when the protein is excited from the ground state or from the O intermediate. Photoexcitation spectra, i.e., FTIR difference spectra directly recorded after flash activation, provide insight into such alterations. A comparison of the excitation spectra of GtACR1 WT after one, two, and five flashes, as well as with continuous exposure, reveals changes in the positive bands (see Fig. 5d). The positive bands in these spectra provide information about the reached intermediate, while the negative bands reflect changes in the ground state or the O intermediate respectively.

With increasing light exposure, band shifts are observed from 1659 cm$^{-1}$ to 1667 cm$^{-1}$ (see Fig. 5a), from 1555 cm$^{-1}$ to 1562 cm$^{-1}$ (see Fig. 5b), and from 1184 cm$^{-1}$ to 1176 cm$^{-1}$ (see Fig. 5c). As previously described, the bands at 1659 cm$^{-1}$ and 1555 cm$^{-1}$ indicate changes in the backbone of the protein and surrounding the retinal. The 1184 cm$^{-1}$ band is a marker for the protonated 13-cis configuration of the retinal. Regarding the negative bands, a shift from 1645 cm$^{-1}$ to 1650 cm$^{-1}$ is observed.

Although shifts of 4 cm$^{-1}$ to 8 cm$^{-1}$ occur, these are comparatively small for IR shifts, which is why it can be assumed that GtACR1 enters a related photocycle upon excitation of the O intermediate. The bands most affected are those that provide information about the global structure of the protein, particularly those in the amide I and II regions. However, it is important to note that the increased band shifting observed under repeated exposure in the amplitude spectra is also present in the excitation spectra.

Of particular importance is the following observation, illustrated in Fig. 5a: The negative band at 1708 cm$^{-1}$, which represents the deprotonation of E68, diminishes under increased illumination conditions, consistent with the lack of reprotonation observed in the amplitude spectra of the corresponding photocycle. This indicates that E68 remains deprotonated throughout these conditions.

The band at 1691 cm$^{-1}$ is associated with conformational changes during pore formation. The decrease in this band under strong illumination, where efficient ion current is observed, suggests that the pore is not fully closed upon excitation from the O intermediate and therefore does not need to open at the beginning of the shortened photocycle. These two band changes are fully consistent with the previously described amplitude spectra, supporting the idea that E68 remains deprotonated under repeated excitation and that secondary structural changes occur during pore formation. Other retinal-related bands, such as 1184 cm$^{-1}$ (assigned to the C–C bond) and 1555 cm$^{-1}$ (assigned to the C=C bond), show slight shifts, indicating changes in the environment of the retinal.

These findings further support our hypothesis that the changes induced by the deprotonation of E68, which contribute to pore formation in GtACR1, are pre-formed upon repeated excitation. This strengthens our postulate of a "pre-opened" state in GtACR1, which, under continuous illumination, stabilizes and facilitates rapid channel opening. This process significantly contributes to the effectiveness of GtACR1 and helps explain the consistently high photocurrents observed in electrophysiological experiments[48].

The findings also apply to the GtACR1 variant Q46E. The high attenuation of the photocurrent observed in this variant, compared to the WT, can be attributed to the prolonged presence of the $L_2$ and M intermediates. With the decay of the $L_2$ intermediate and the formation of the M intermediate, the channel enters a non-conductive state, and since the retinal remains in the 13-cis configuration, no new excitation can occur. Once the O intermediate is reached, a photocycle is also initiated in the Q46E variant (see Supplementary Fig. 2).

### Extended photocycle model

Our data allows us to expand the photochemical model of GtACR1 (see Fig. 6), as explained in detail below. According to our current view, the initial stages of the photocycle, as outlined in the introduction, remain unchanged. Photoexcitation ($\lambda$ = 480–500 nm) of the ground state ACR1 leads to retinal isomerization, initiating the K intermediate, followed by the non-conducting $L_1/L_1'$ intermediate and the conducting $L_2$ intermediate. Subsequently, channel closing occurs in two steps: from $L_2$ to M and from M to

**Fig. 5 | Comparison of the excitation spectra after different illumination times.** The respective excitation spectra after one (black), two (red), and five (blue) flashes, as well as after continuous exposure to light (green), are shown (**d**). A shift can be observed with increasing exposure from 1659 cm$^{-1}$ to 1667 cm$^{-1}$ (**a**), from 1555 cm$^{-1}$ to 1562 cm$^{-1}$ (**b**), and from 1184 cm$^{-1}$ to 1176 cm$^{-1}$ (**c**). 1659 cm$^{-1}$ and 1555 cm$^{-1}$ represent the amide region of the proteins. 1184 cm$^{-1}$ is a marker band for protonated 13-*cis* retinal.

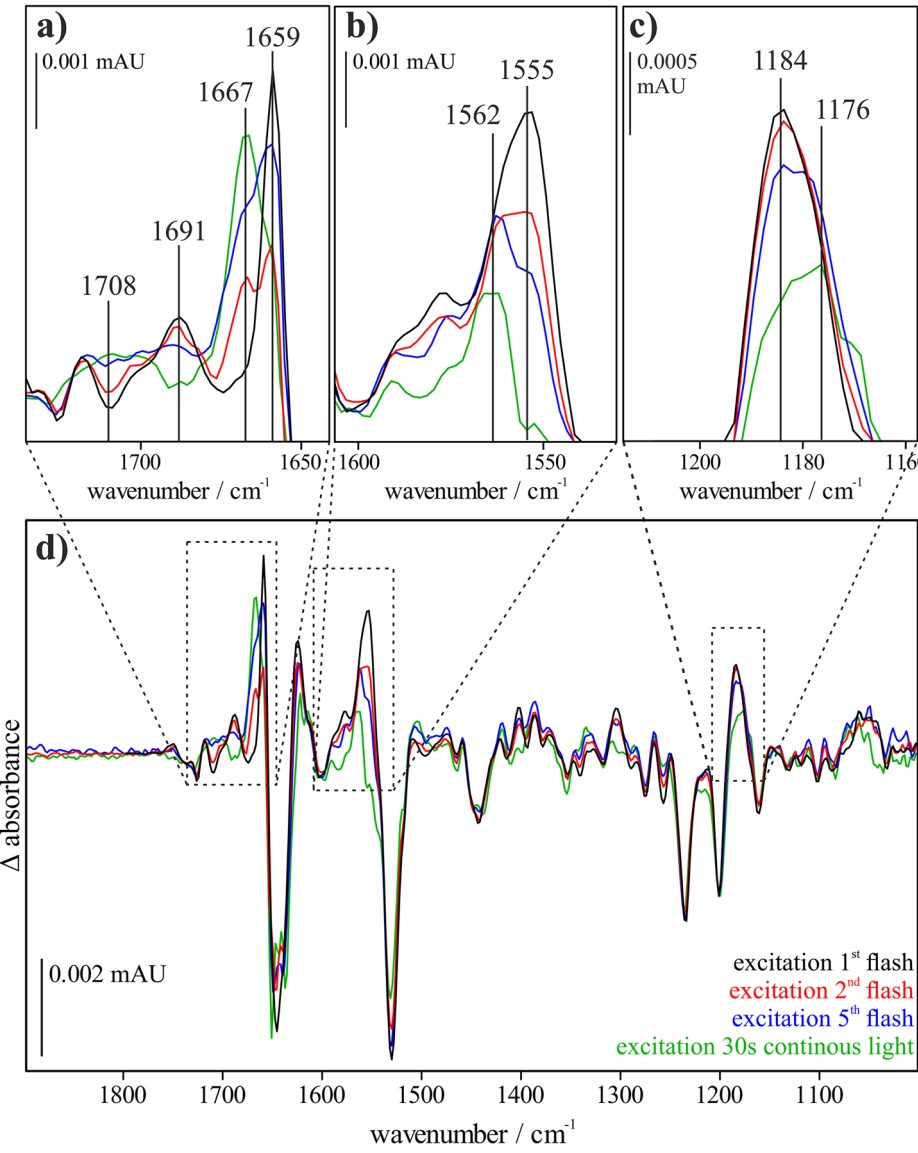

N/O. In the previous version of the cycle, *Gt*ACR1 relaxes into the ground state from the N/O intermediate with a half-life of 4.4 s, a process that still occurs without further photoexcitation.

However, photoactivation of the O intermediate, which contains all-*trans* retinal and the pore in a "pre-open" state, generates a shortened photocycle. The first observable intermediate within our time resolution is L$_2$; therefore, we cannot characterize earlier intermediates. We hypothesize that with the transition to the 13-*cis* configuration, a short-lived K-like intermediate, which we refer to as K', is reached before entering the known photocycle at the L$_1$/L$_1$' intermediate, followed by channel opening.

The identification of the O intermediate as a photoactivatable state provides the opportunity to reconcile the apparently conflicting data from optical and FTIR spectroscopy using a flash activation setup, as well as electrophysiological measurements based on continuous illumination (see Fig. 2b).

According to our hypothesis, *Gt*ACR1 enters a highly conductive photocycle upon initial excitation, which is reflected in a high current deflection in the photocurrent. *Gt*ACR1 undergoes the well-known photocycle described by Dreier et al.[66] and returns to the ground state in the event of single-flash activation. In the case of continuous exposure, as in the electrophysiological measurements[48,51,74], steady state conditions are reached after the initial peak of the first photocycle. The amount of attenuation is determined by the ratio of the half-lives of the open-state intermediates and the non-excitable closed-state intermediates. The key is the M intermediate, which has a closed state and is not excitable. The half-life of M is about 10 times longer for the Q46E mutant, resulting in about 10 times stronger attenuation.

These insights offer a new perspective on the previously published data by Szundi et al.[87,88], proposing a parallel photocycle kinetic model and a red-absorbing intermediate as the open channel state. Furthermore, the hypothesis that the O intermediate, retaining an all-*trans* retinal configuration and a "pre-open" state of the central gate, might allow for rapid channel reopening could provide a new framework for understanding sustained ion conduction under continuous illumination. A thorough understanding of the gating mechanisms and attenuation in light-activated ion channels is pivotal for the rational design of next-generation optogenetic tools.

## Conclusion

During our investigations of the *Gt*ACR1 variant Q46E, in an effort to gain further insights into the strong attenuation effect associated with the *syn*-photocycle, we observed the absence of the typical marker band for 13-*cis* retinal in *syn*-configuration, as well as significantly retarded partial reaction rates, leading to an enrichment of non-conducting intermediates.

**Fig. 6 | Revised photocycle model of *Gt*ACR1.** The revised photocycle model is based on previously published FTIR- and UV/Vis-spectroscopic data combined with published electrophysiological measurements and our new ambient temperature time-resolved FTIR-spectroscopic data. It explains the efficient ion conductance despite the long-lived late intermediates with a closed channel by the possibility to excite the O intermediate, leading to retinal isomerization from all-*trans* to the 13-*cis* C=N-*anti* configuration. The first observable intermediate within our time resolution is $L_2$; therefore, we cannot characterize earlier intermediates. We hypothesize that with the transition to the 13-*cis* configuration, a short-lived K-like intermediate, which we refer to as K', is reached before entering the known photocycle at the $L_1/L_1'$ intermediate, followed by channel opening. As this intermediate is not experimentally measured, it is represented by a dotted box.

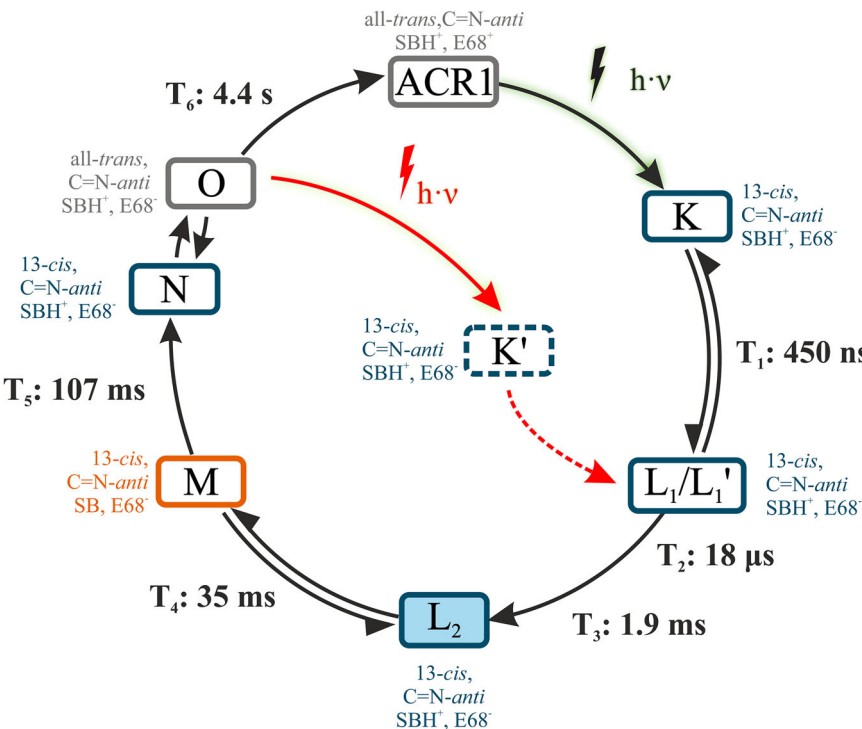

This raised the question of the wild-type kinetic behavior, prompting us to investigate distinct illumination schemes. We found that with the arrival of the all-*trans* configuration of retinal at the O intermediate, another round of the photocycle can be triggered by flash irradiation. Amplitude spectra derived from global fit analysis, along with photoexcitation spectra, showed a high degree of resemblance between the O intermediate-generated and the ground-state-generated courses, suggesting that their functional behavior is closely related.

Subtle alterations in the evolution of secondary structural changes and protonation events within the central constriction site of the channel—such as changes in the protonation of E68 or signs of pore formation—led us to propose a so-called "pre-open" state, which facilitates easier channel opening.

Our investigations of *Gt*ACR1 have revealed that the high efficiency of the channel is due to the second photoactivatable state, which allows the photocycle to be re-excited from the O intermediate and is associated with the formation of a "pre-open" state.

This finding offers new approaches for improving *Cr*ChR2 for optogenetic applications, aiming to increase the channel's conductivity.

## Methods
### Site-specific mutagenesis and preparation of mutant DNA
If not indicated otherwise, all molecular biological work was carried out as described in Dreier et al.[66]. To obtain the Q46E variant, we used the site-specific mutagenesis Quik-change protocol using the template plasmid pPIC9k-ACR1-His10 and the oligonucleotide primers GTTGT TTCTGCTTGT<u>GAA</u>GTTTTCTTTATGG and CCATAAAGAAAACTT CAC<u>AAG</u>CAGA<u>AA</u>CAAC.

### Transformation of Pichia pastoris, mutant selection, and ACR1 expression
*Pichia pastoris* strain SMD 1168 (Thermo Fisher Scientific, MA) is streaked onto a YPD agar plate and incubated for 72 h at 30 °C. Subsequently, 10 mL of YPD medium is inoculated with a single colony from the YPD plate and shaken overnight at 30 °C and 150 rpm. The next day, 50 mL of YPD medium is inoculated with the 10 mL culture to an $OD_{600}$ of 0.2 and shaken at 30 °C and 150 rpm until an $OD_{600}$ of 1–1.6 is reached. After centrifugation at $1500 \times g$ and 4 °C for 5 min, the resulting sediment is resuspended in 50 mL sterile ice-cold water and centrifuged again under the same conditions. The sediment is resuspended in 25 mL sterile ice-cold water, followed by another centrifugation at the same settings. Finally, the sediment is resuspended in 2 mL of 1 M sterile ice-cold sorbitol and centrifuged again as previously described. After removal of the supernatant, the cells are taken up in 100 µL sterile ice-cold sorbitol. Subsequently, 12 µg of PmeI linearized DNA is added to 80 µL of the cells, incubated on ice for 5 min, and transferred using electroporation at 1.5 kV, 25 µF, and 200 Ω. Immediately afterwards, 1 mL of ice-cold 1 M sorbitol is added to the mixture, which is then shaken for 60 min at 30 °C and 450 rpm. The transformed cells, now containing expression vectors with HIS4 genes, are streaked out on an MD plate and incubated at 30 °C for 72 h to select for the uptake of the plasmid. Colonies were picked and transferred to plates containing the selectant Geneticin at increasing concentrations (0.25 mg/mL, 0.5 mg/Ll, 0.75 mg/mL, and 1 mg/mL) and incubated at 30 °C for 1 week to check for the number of copies. Thereafter, 6 precultures are prepared by inoculation of 50 mL BMGY, with a colony that survived the highest geneticin concentration, and shaken at 30 °C and 170 rpm. By use of PCR from precultures, positive clones carrying the desired gene fragment are verified and used to inoculate 1 L of BMGY medium[89]. The culture is shaken at 30 °C and 100 rpm overnight. To prepare the main culture, $6 \times 1$ L of BMGY medium is inoculated with the preculture to a starting $OD_{600}$ of 1. Furthermore, 10 µL of a stock solution all-*trans* retinal was added to 10 mL of 100% methanol, and the mixture was added to each flask[90,91]. The cultures are shaken at 30 °C and 90 rpm. After 6 h, 5 µL all-*trans* retinal is added to 5 mL 100% methanol. After 24 h, 10 µL all-*trans* retinal in 10 mL 100% methanol is added again as at the beginning, and after 48 h, 5 µL all-*trans* retinal in 5 mL 100% methanol is added. After shaking for another 6 h, the cultures are harvested for 15 min at 5000 rpm and 15 °C in an Avanti centrifuge. The sediments are resuspended in 20 mL lysis buffer (50 mM sodium phosphate, pH 7,5, 5% glycerol, 1 mM EDTA, 500 mM NaCl, 1 mM PMSF) and stored at −20 °C.

### Membrane preparation and protein purification
Cells were disrupted by 15 cycles at 1500 bar using a microfluidizer M-110L (Microfluidics Corp., Newton, MA) and lysis buffer. Non-disrupted cells

and large debris are removed by centrifugation for 15 min at 5000 rpm and 4 °C. The membrane containing supernatant is sedimented by ultra-centrifugation for 1 h 15 min at 45,000 rpm and 4 °C. The obtained membrane components are homogenized in a 9-fold amount of solubilization buffer (20 mM HEPES, pH 7.5, 500 mM NaCl, 1 mM PMSF, 1% (w/v) n-Dodecyl-beta-maltoside) and solubilized overnight at 4 °C under stirring in the dark or under red light. After the addition of 50 mM imidazole, another ultracentrifugation (1 h 15 min at 45,000 rpm and 4 °C) is performed to remove unsolubilized membranes, and the supernatant is used for affinity chromatography. *Gt*ACR1 purification was performed by Ni-NTA affinity chromatography using a HisTrap™ *Fast Flow* 5 mL column (Cytiva, Dassel, Germany) and a step gradient elution (buffer A: 20 mM Hepes, 100 mM NaCl, 0,15% DM, pH 7.5; buffer B: 20 mM Hepes, 100 mM NaCl, 0,15% DM, 500 mM imidazole, pH 7.5). Colored fractions are pooled and loaded on a gel filtration column using a HiLoad 16/600 Superdex 200 pg (GE Healthcare, Düsseldorf, Germany), and eluted with buffer A, yielding purified *Gt*ACR1, which is characterized by SDS-Page and Western blotting.

### Reconstitution of *Gt*ACR1 using egg phosphatidylcholine and preparation of samples for FTIR spectroscopy

To simulate conditions of a natural lipid environment[92] in spectroscopic measurements, purified *Gt*ACR1 are reconstituted into egg phosphatidylcholine (Avanti Polar Lipids, AL). The lipids are solubilized with 0.15% n-Dodecyl-beta-maltoside in 20 mM HEPES, pH 7.5, 100 mM NaCl by incubation at 50 °C for 10 min. Solubilized lipids and purified *Gt*ACR1 are mixed at a 2:1 ratio (w/w, lipid to protein) and incubated for 30 min. The detergent is removed overnight by adsorption on Bio-Beads SM 2 (BioRad, CA) using 40:1 (w/w, Bio-Beads to detergent) at room temperature. The following day, this procedure of detergent removal is repeated for 4 h. The resulting suspension containing proteoliposomes and buffer is further processed by separating the proteoliposome suspension from the beads and centrifuging for 10 min at 20,000 rpm. The resulting pellet is then resuspended in measuring buffer (20 mM HEPES, 100 mM NaCl, pH 7.5) and ultracentrifuged at 55,000 rpm for 3 h using a Thermo Scientific MTX150 micro-ultracentrifuge with a S55-A2 rotor. To assemble the sample, two CaF$_2$ windows (∅ 2 cm, 2 mm thickness, one of them with a 10 μm deepened area 1 cm in diameter) were cleaned with detergent, isopropyl alcohol, and de-ionized water. The edge of one window was greased with a thin layer of silicon grease to seal the sample. The pelleted protein/phospholipid sample containing either wild-type or mutant full-length *Gt*ACR1 is applied to the deepened window and covered by the second window to obtain an optical path length between 5 and 10 μm. The double window stack containing samples is sealed, placed in a metal cuvette, and mounted to the FTIR spectrometer (Bruker Vertex 80 v, Bruker Corporation, MA, USA) at 21 °C. Samples were equilibrated overnight.

### FTIR-experiments

Time-resolved FTIR difference spectroscopy was performed to gain insight into the changes upon illumination. After sample equilibration, background spectra were taken (400 scans) and the samples were illuminated with a short laser pulse of a Minilite Nd:YAG laser (Continuum, Pessac, France, $\lambda_{max}$: 532 nm, 12 ns pulse) in case of single- and multi-flash measurements. For continuous illumination measurements, green LED lights were used ($\lambda_{max}$: 525 nm). Measurements were performed in the rapid-scan mode of a Vertex 80 v spectrometer and OPUS 7.2 software (Bruker Corporation), an Adwin Pro II A/D converter and ADbasic 6 software (Jäger Computergesteuerte Messtechnik GmbH), and a Lecroy WaveRunner HRO64zi oscilloscope with WaveRunner 6 Zi Oscilloscope Firmware version 6.6.0.5 (Teledyne LeCroy) and MatLab R2015a (The Math-Works, Inc.). Data between 1900 and 1000 cm$^{-1}$ were collected with a spectral resolution of 4 cm$^{-1}$ in the double-sided forward-backward data acquisition mode with a scanner speed of 120 kHz. For the Fourier transformation, a zero-filling factor of 4 and Norton-Beer weak apodization was applied.

### Global fit analysis

For the analysis of the time-resolved data, a global fit[84] in MATLAB (The MathWorks, MA, USA) was used[84], as described by Chizow et al.[81–83]. The time-resolved absorbance change $\Delta A$ (v, t) of *Gt*ACR1 measurements is described by the absorbance change induced by photoactivation $a_0(v)$ followed by three exponential functions fitting the amplitudes $a$ for each wavenumber $v$ (Eq. 1).

$$\triangle A(v,t) = a_0(v) + a_1(v)\left(1 - e^{-k_1 t}\right) + a_2(v)\left(1 - e^{-k_2 t}\right) + a_3(v)\left(1 - e^{-k_3 t}\right)$$

Dreier et al. identified $a_1(v, t)$ as the transition from L$_2$ to M, $a_2(v, t)$ as transition from M to N/O and $a_3(v, t)$ as transition from N/O to the ground state ACR1[66]. In the figures, disappearing bands face upward and appearing bands face downward. Rapid-Scan spectra were obtained from a total of 48 measurements from three protein samples per measurement condition.

### Statistics and reproducibility

Rapid-scan spectra were obtained from a total of 48 measurements from three protein samples per measurement condition. The number of repetitions was adjusted according to data quality. IR-Measurements with large baseline drifts were excluded.

### Reporting summary

Further information on research design is available in the Nature Portfolio Reporting Summary linked to this article.

### Data availability

The raw data of the spectroscopic measurements, as well as the simulation trajectories and input files, will be provided upon individual request by the corresponding authors. Additionally, all source data underlying the kinetics and spectra presented in the main figures can be found in Supplementary Data 1.

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

## Acknowledgements

We thank Elena Govorunova for providing the raw electrophysiological data for Fig. 1a. We thank Max-Aylmer Dreier for all the support and helpful discussions, as well as Simon Völker and Olga Zapolskaia, who contributed to the project during their respective thesis. We acknowledge Harald Chorongiewski for his technical support with the spectroscopic setup and Gabriele Smuda for molecular biology support. This work was supported by Deutsche Forschungsgemeinschaft (DFG, German Research Foundation) Individual Research Grant, "Molecular mechanisms of cation and anion-conducting channelrhodopsins" (GE 599/23-1) to K.G. and the DFG Priority Program SPP1926 (GE 599/19-2 and GE 599/19-1) to K.G. Further support was provided by the Ministry for Culture and Science (MKW) of North-Rhine-Westphalia (Germany) through grant 111.08.03.05-133974 to K.G. and the Protein Research Unit Ruhr within Europe (PURE) funded by the Ministry of Innovation, Science and Research (MIWF) of North-Rhine-Westphalia (Germany) to K.G.

## Author contributions

K.G. obtained the funding; K.G., M.L., T.R., and C.K. designed the research; K.L., M.J.N., L.-M.H., and P.A. performed the research; K.L. and M.J.N. performed FTIR measurements with the help of L.-M.H., supervised by K.G., T.R., and C.K.; K.L. and L.-M.H. performed biochemistry supervised by M.L.; all authors analyzed the data; K.L., K.G., M.L., T.R., and C.K. wrote the paper with edits from all co-authors.

## Funding

## Competing interests

The authors declare no competing interests.
