## [Transparent Peer Review file · Communications Biology]

A Second Photoactivatable State of the Anion-conducting channelrhodopsin GtACR1 empowers persistent activity

Corresponding Author: Dr Carsten Kötting

Version 0:

Reviewer comments:

Reviewer #1

(Remarks to the Author)

Channelrhodopsins (ChRs) are light-gated ion channels widely utilized in optogenetics for the optical control of neuronal activity. However, the most widely used ChR, CrChR2, exhibits photocurrent attenuation upon prolonged illumination. By contrast, GtACR1, a recently discovered anion channelrhodopsin, does not show such photocurrent attenuation. In this study, the authors investigated the gating mechanism of GtACR1, which differs from that of CrChR2, using time-resolved FTIR spectroscopy. Their results suggest that photoexcitation of the red-shifted O-intermediate leads to the formation of a pre-open state, which facilitates rapid channel opening and reduces attenuation during prolonged illumination. Additionally, they propose that the protonation and deprotonation of E68, a residue highly conserved among ChRs including CrChR2, is essential for this process.

However, without electrophysiological measurements performed under similar light illumination conditions, it is difficult to conclude that O-intermediate excitation indeed directly leads to the formation of a pre-open state facilitating strong photocurrents upon prolonged illumination. Given that the authors' conclusions rely solely on spectroscopic data, additional evidence is needed to fully support their claims. As a result, the study in its current form may not appeal to a broad readership appropriate for *Communications Biology*. I recommend considering submission to a more specialized journal. Nevertheless, to improve the manuscript, I outline several major concerns that should be addressed, along with some minor points.

[Major Points]

Page 5, Lines 173-175:

"Similarly, it was calculated that the GtACR1 ground state is populated with only a negligible 4 % after 1 second of activation, with 96 % of the proteins remaining in the N/O intermediate."

My primary concern is that this calculation assumes that 100% of the GtACR1 proteins in the illuminated area were converted into photointermediates upon the first flash. However, for rhodopsins, this is generally difficult due to the quantum yield of all-trans-to-13-cis retinal isomerization significantly smaller than 100% and the back-isomerization reaction from the K-intermediate to the initial state within the nanosecond excitation pulse (e.g., Malakar et al., *J. Phys. Chem. Lett.* (2022) 13, 8134-8140). The fraction of unreacted proteins after the first flash should be carefully estimated. Furthermore, if a significant amount of GtACR1 remains in the ground state after the first flash, its contribution to the FTIR spectra shown in Figures 3 and 4 must be taken into account.

Page 3, Line 77:

"K decays to the non-conducting L1/L1' intermediate."

The most recent transient absorption measurements in the visible region (Refs. 50 and 51) indicate that the O-intermediate is involved in the conductive state of GtACR1. However, this finding is not clearly reflected in the mechanistic models presented in this work. This point should be addressed, and if necessary, the mechanistic models and Figures 2 and 6 should be revised accordingly.

Page 6, Lines 218-221:

"This interpretation is further supported by the increased formation of the band at 1691 cm^{-1} during stronger light exposure (see Figure 4f/g), which reflects alterations in the secondary structure during channel opening and clearly supports the buildup of the postulated pre-open state of the ion channel."

If 30 seconds of illumination enhances the 1691 cm^{-1} peak representing the channel opening, one might expect that longer

exposure would further open the channel and increase the photocurrent. However, the stationary current is weaker than the peak current, as shown in Figure 1b. Please clarify this apparent inconsistency.

Minor Points

Page 4, Lines 132-133:

"Channel opening itself and the conducting state, visible at 1691 cm^{-1} , follow the same pattern as described in literature for the wild type."

The strong similarity of the 1691 cm^{-1} peaks, which represent the channel-open state, between GtACR1 WT and Q46E seems inconsistent with the large difference in peak current observed in Figure 1b. Please clarify why the Q46E mutant, which exhibits a similar channel-opened structure to the WT, has a much smaller peak photocurrent.

Page 7, Lines 239-240:

"...as the positive bands exhibit only minor changes."

In Figure 5, many significant changes are observed in the positive bands. Is this a typographical error, and should it be "negative bands"?

Page 7, Lines 251-252:

"... the pore is not fully closed and therefore does not need to open at the beginning of the photocycle."

If the pore is not fully closed, a leak current, whose sign should change depending on the holding potential during electrophysiological measurements, would be expected even before the photocycle begins. However, to the best of my knowledge, such a leak current has not been observed for GtACR1, except for the E68R mutant (Sineshchekov et al., PNAS (2015), 112, 14236-14241). Please clarify this point.

Page 8, Lines 264-265:

"... the L2 and M intermediates. During these states, the pore is in a non-conductive state, and..."

Does L2 not represent a conductive state, as shown in Figure 2?

Page 8, Lines 295-296:

"... new approaches for improving CrChR2 for optogenetic applications, aiming to increase the channel's conductivity, can be found based on these findings."

It is unclear what specific approaches could be derived from the findings of this study to improve CrChR2 for optogenetic applications. Please provide a more concrete explanation.

Page 10, Lines 351-352:

"In addition, 10 μl all-trans retinal in 10 ml 100 % methanol is added to each flask."

The meaning of "10 μl all-trans retinal in 10 ml 100% methanol" is unclear. Does this mean that 10 μl of a stock all-trans retinal solution was added to 10 ml of methanol? Please rephrase this sentence for clarity.

Page 11, Lines 368 and 384:

"maltosid" should be corrected to "maltoside."

Figure 1b:

The photocurrent intensity of ChRs can vary depending on cell type, ion composition, membrane potential, light conditions, etc. Were the photocurrents shown in Figure 1b recorded under identical experimental conditions? Otherwise, comparing their absolute intensities may not be meaningful. If all measurements were conducted under the same conditions, please specify the experimental details.

Figures 3c and 4a, and Supplementary Figure 2:

Labeling the prominent peaks would enhance readability.

Figure 5:

How were these excitation spectra obtained? Please describe the experimental details used to acquire these spectra.

Figure 6:

Indicating the protonation state of E68 in each intermediate would help clarify the proposed mechanism.

Reviewer #2

(Remarks to the Author)

The authors present the discovery of a novel secondary-photon-activation photochemical mechanism that greatly increases the conductance by the light-gated anion channelrhodopsin GtACR1. Their results are compelling, and the second-photon photochemical reactions are revealed by their innovative experiments and their expertise with time-resolved kinetic FTIR, and their comparison of GtACR1 and the cation channel CrChR2 structures and photochemical reactions, two widely used channelrhodopsins as optogenetic tools. The enhancement of ion currents through the membrane by the mechanism they unraveled explains the much greater conductance of GtACR1 in continuous light compared to that of CrChR2 in the same condition.

The results presented are of immediate relevance to researchers applying optogenetics (e.g. in neuroscience or cardiac cell

research, which involve channelrhodopsins). The discovery of this new mechanism also establishes a new concept to be considered by investigators of any photoactivated protein that undergoes cyclic photochemical reactions (photocycles), therefore the impact of this paper reaches also a broad audience in photochemistry and photobiology. In continuous light, secondary photochemistry occurs in GtACR1, which entails absorption of a second photon by a late photocycle intermediate that drives the protein back to an early photocycle intermediate conformation. This early intermediate quickly converts to the conformation that opens the channel. Therefore, the authors have discovered an elegant, novel mechanism produced by evolution that avoids disrupting, but rather enhances, the effective single-photon photocycle by creating a second-photon photocycle within the single-photon photocycle. The result is that the photochemical reactions produce a longer lasting conducting open-channel conformation in the protein.

The writing is concise and well written. The research is clearly and accurately described both in the history of channelrhodopsins and the new exciting discovery.

Reviewer #3

(Remarks to the Author)

The manuscript under review addresses an important issue in optogenetics—the phenomenon of photocurrent attenuation in anion-conducting channelrhodopsins—and seeks to elucidate its molecular basis by focusing on GtACR1 and a Q46E mutant variant. The authors present a comprehensive FTIR spectroscopic investigation that attempts to correlate subtle changes in retinal configuration and protein conformational dynamics with the functional phenomenon of attenuation. This is a timely subject, given that a thorough understanding of the gating mechanisms and attenuation in light-activated ion channels is pivotal for the rational design of next-generation optogenetic tools.

The work is conceptually motivated by the challenge of overcoming the attenuation observed in cation channelrhodopsins such as CrChR2. By comparing GtACR1 with its mutant Q46E, the authors draw attention to the role of the central gate—a region that is conserved between ChR2 and ACR1—in modulating ion conductance and the kinetics of channel closing. The manuscript carefully documents the alterations in photocycle reaction rates in the mutant relative to the wild type, and it proposes that the accumulation of intermediates preceding the photoactivatable ground state is responsible for the observed attenuation. The hypothesis that the O intermediate, retaining an all-trans retinal configuration and a “pre-open” state of the central gate, might allow for rapid channel reopening is intriguing and, if substated, could provide a new framework for understanding sustained ion conduction under continuous illumination.

Despite the conceptual clarity and logical progression in the authors’ argument, several critical issues remain that require further attention, therefore I would recommend major revision of the manuscript. The experimental design relies exclusively on FTIR spectroscopy to probe the dynamics of the photocycle and does not include complementary electrophysiological measurements. In previous studies on CrChR2, a combination of FTIR, electrophysiology, and Raman spectroscopy was employed to correlate spectral changes with functional currents (Kuhne et al PNAS, 2019, doi.org/10.1073/pnas.1818707116). The absence of such direct functional data in the current study limits the ability to unambiguously link the spectroscopic signatures to channel conductance, particularly regarding the proposed rapid re-excitation of the O intermediate. Direct evidence, such as the dependence of photocurrent magnitude on illumination intensity or pH, would greatly strengthen the authors’ claims regarding the mechanistic origin of attenuation.

The choice of mutation in the central gate also raises some concerns. The substitution of a glutamine with a glutamate in the Q46E mutant is interesting, yet the rationale behind this specific change is not fully justified, especially in light of the observation that the mutant still exhibits a similar attenuation phenomenon. One would expect that replacing a neutral residue with a charged one could significantly alter local electrostatics, and therefore, a clear dependence on pH should be evident. However, the manuscript does not specify the pH conditions under which the FTIR experiments were performed, leaving an important experimental variable unaddressed. A more detailed discussion of the pH sensitivity of the system is warranted, as the protonation states of residues such as E68 are central to the proposed model of the “pre-open” state.

Minor comments:

Line 15 “Optogenetics is a method to regulate cells using light.” Possibly, it might be better to mention not only cells, but also tissues and organisms.

Line 16 It is appropriate bet to mention not only neurons but also other targets of optogenetics

Line 17 Might I suggest changing “very low” to “lower”?

Line 18 Could you explain what do you mean a “branched photocycle”? Could it be two different photocycles of different fractions of rhodopsins (like in ChR2 (Kuhne et al PNAS, 2019, doi.org/10.1073/pnas.1818707116))?

It would be rational to mention in the Introduction other examples of photoswitchable photocycle intermediates for other rhodopsins, for example for bacteriorhodopsin.

Line 124 During describing the procedure of global fit analysis it would be correct to mention original works by Chizov et. al about this method (Chizhov, I. et al., Biophysical Journal, Volume 71, Issue 5, 2329 – 2345)

Line 220 In the sentence “alterations in the secondary structure during channel opening and clearly supports the buildup of the postulated pre-open state of the ion channel.” Instead of using the word “clear”, please, explain why it “supports”?

Line 278 Why this intermediate is called “pre-opened” state? For example, for CrChR2 it was shown that opening of the channel for protons occurs much earlier in the photocycle (during M intermediate), and O intermediate is the last one in the photocycle. Maybe this intermediate could be called “pre-closed” state?

In Fig.6 you represent a photocycle model with K-intermediates, but as I understood, experimental data in this work was obtained starting from the L-intermediate (due to time resolution limits). Could you, please, mark also in the Fig.6 that a short-

lived K-like intermediate is hypothetical (not directly measured experimentally).

Version 1:

Reviewer comments:

Reviewer #1

(Remarks to the Author)

Although the authors have revised many parts of the manuscript in response to my comments, unfortunately, two major concerns remain insufficiently addressed.

Regarding the request from both Reviewer 3 and myself for electrophysiological experiments to support the conclusions drawn from FTIR data, the authors explained the difficulties in performing such measurements and instead added a detailed explanation of the photocurrent data in Figure 1, referencing the original studies. However, while the photocurrents in the study by Govorunova et al. were measured at a holding potential of -60 mV in HEK293FT cells 10 to 19 days after transfection, Kim et al. performed patch-clamp recordings in HEK293 cells 24–48 hours after transfection. Therefore, the protein environments differ considerably, making normalization using the GtACR1 wildtype data difficult. In fact, in the latter case, the negative photocurrent shows strong attenuation, which was not observed under the experimental conditions used in the former study. Thus, reproducing the data from these studies in Figure 1 is not appropriate and misleading for the readers. The photocurrents of the three proteins should be recorded under the same experimental conditions, and an electrophysiological study supporting the authors' conclusions is necessary. If this is difficult using the authors' current system, they could consider collaborating with another group equipped with the appropriate instrumentation for the purposes of this study. Although the authors state that their light illumination pattern is difficult to reproduce under electrophysiological conditions, many electrophysiology laboratories are capable of using nanosecond 532-nm laser pulses or 525-nm LEDs as stimulation light sources. Therefore, this does not justify avoiding such collaboration. Otherwise, since the relationship between the FTIR-detected structural changes and the channel-opening mechanism cannot be fully understood without electrophysiological data, I recommend submitting the manuscript to a more specialized journal.

In Answers to the Reviewers:

"...we performed lower intensity experiments at the beginning of the studies to ensure that we are measuring in saturation. We measured the samples at half the laser intensity used in the data shown, resulting in comparable excitation amplitudes indicating that the yield of excited proteins is indeed in saturation... Furthermore, reviewer 1 is correct that intensity and illumination time are crucial factors of how many proteins are converted into photointermediates. However, the referred paper by Malakar et al. used a very short laser pulse with 100 fs. In our experiments, we used a 12 ns pulse (120-fold longer). The lifetime of the retinal excited state is about 500 fs. Thus, proteins that are return to the ground state without isomerization can be excited again to enter the photocycle"

This point seems different from the one I raised. My concern is the suppression of photoproduct accumulation due to back-isomerization from the K intermediate (13-cis) to the initial all-trans state, not from the retinal excited state. This back-isomerization also occurs when a nanosecond laser pulse is used as the excitation source. Signal saturation does not guarantee 100% accumulation of the photoproduct. During strong laser pulse illumination, a photo-equilibrium between the initial and K-intermediate states is established. The ratio between these states is governed by the quantum yields of all-trans to 13-cis isomerization (from the initial state) and 13-cis to all-trans isomerization (from the K intermediate), the latter being higher than the former as reported in Malakar et al., *J. Phys. Chem. Lett.* (2022) 13, 8134–8140, as well as by their respective extinction coefficients at the excitation wavelength. Please take these considerations into account, and I recommend the authors carefully evaluate the populations of photointermediates.

Reviewer #3

(Remarks to the Author)

I carefully read the revised manuscript and the answers of the authors, I am quite satisfied with their revision and I think that this manuscript can be useful for scientific community. I recommend it for publication.

Version 2:

Reviewer comments:

Reviewer #1

(Remarks to the Author)

The authors appropriately addressed my comments. I recommend publishing this manuscript in *Communications Biology*.

Answers to the Reviewers

Reviewer #1 (Remarks to the Author):

Channelrhodopsins (ChRs) are light-gated ion channels widely utilized in optogenetics for the optical control of neuronal activity. However, the most widely used ChR, CrChR2, exhibits photocurrent attenuation upon prolonged illumination. By contrast, GtACR1, a recently discovered anion channelrhodopsin, does not show such photocurrent attenuation. In this study, the authors investigated the gating mechanism of GtACR1, which differs from that of CrChR2, using time-resolved FTIR spectroscopy. Their results suggest that photoexcitation of the red-shifted O-intermediate leads to the formation of a pre-open state, which facilitates rapid channel opening and reduces attenuation during prolonged illumination. Additionally, they propose that the protonation and deprotonation of E68, a residue highly conserved among ChRs including CrChR2, is essential for this process.

However, without electrophysiological measurements performed under similar light illumination conditions, it is difficult to conclude that O-intermediate excitation indeed directly leads to the formation of a pre-open state facilitating strong photocurrents upon prolonged illumination. Given that the authors' conclusions rely solely on spectroscopic data, additional evidence is needed to fully support their claims. As a result, the study in its current form may not appeal to a broad readership appropriate for Communications Biology. I recommend considering submission to a more specialized journal. Nevertheless, to improve the manuscript, I outline several major concerns that should be addressed, along with some minor points.

We understand the concerns of Reviewer 1 and agree that electrophysiological measurements under similar light conditions is scientifically interesting. Unfortunately, we do not have the necessary equipment in our laboratory to carry out these measurements. Replicating similar lighting conditions in another laboratory is very complicated due to the complexity of the laser we use. However, electrophysiology measurements of all protein variants covered in our manuscript are available and discussed within the manuscript. Our spectroscopic data is in full agreement with these electrophysiological measurements. Furthermore, we included FTIR measurements with illumination conditions similar to the electrophysiological measurement, in order to show that the processes are the same. The results nicely explain all the features of the electrophysiological experiments, including the amount of attenuation and the peak in the beginning. Therefore, the results presented in the manuscript stand for themselves and further electrophysiological experiments go beyond the scope of this manuscript.

The contradictions between the electrophysiology and our measurements in both the WT and the Q46E mutant mentioned by reviewer 1 are based on a misunderstanding and do not exist. We adapted the text in an effort to make this point clearer (see below).

'According to our hypothesis GtACR1 enters a highly conductive photocycle upon initial excitation, which is reflected in a high current deflection in the photocurrent. GtACR1 undergoes the well-known photocycle described by Dreier et al.⁴² and returns to the ground state in the event of single flash activation. In the case of continuous exposure, as in the electrophysiological measurements^{24,27,51}, steady state conditions are reached after the initial peak of the first photocycle. The amount of attenuation is determined by the ratio of the half-lives of the open-state intermediates and the non-excitable closed-state intermediates. The key is the M intermediate, which has a closed state and is not excitable. The half-life of M is about 10 times longer for the Q46E mutant, resulting in about 10 times stronger attenuation'

[Major Points]

Page 5, Lines 173-175:

"Similarly, it was calculated that the GtACR1 ground state is populated with only a negligible 4 % after 1 second of activation, with 96 % of the proteins remaining in the N/O intermediate."

My primary concern is that this calculation assumes that 100% of the GtACR1 proteins in the illuminated area were converted into photointermediates upon the first flash. However, for microbial rhodopsins, this is generally difficult due to the quantum yield of all-trans-to-13-cis retinal isomerization significantly smaller than 100% and the back-isomerization reaction from the K-intermediate to the initial state within the nanosecond excitation pulse (e.g., Malakar et al., J. Phys. Chem. Lett. (2022) 13, 8134-8140). The fraction of unreacted proteins after the first flash should be carefully estimated. Furthermore, if a significant amount of GtACR1 remains in the ground state after the first flash, its contribution to the FTIR spectra shown in Figures 3 and 4 must be taken into account.

The concerns, that our theory is based on the assumption that 100% of the *GtACR1* proteins are excited upon initial excitation are valid. Indeed, we had the same concern at the beginning of our measurement. Therefore, we performed lower intensity experiments at the beginning of the studies to ensure that we are measuring in saturation. We measured the samples at half the laser intensity used in the data shown, resulting in comparable excitation amplitudes indicating that the yield of excited proteins is indeed in saturation. For clarity, we added a statement that measurements were performed under saturation conditions:

'Under our conditions, all molecules are excited. This has been demonstrated by experiments at lower laser intensities, resulting in almost the same absorbance changes. Thus, we are working under saturation conditions.'

Furthermore, reviewer 1 is correct that intensity and illumination time are crucial factors of how many proteins are converted into photointermediates. However, the referred paper by Malakar et al. used a very short laser pulse with 100 fs. In our experiments, we used a 12 ns pulse (120-fold longer). The lifetime of the retinal excited state is about 500 fs. Thus, proteins that are return to the ground state without isomerization can be excited again to enter the photocycle

Page 3, Line 77:

"K decays to the non-conducting L1/L1' intermediate."

The most recent transient absorption measurements in the visible region (Refs. 50 and 51) indicate that the O-intermediate is involved in the conductive state of GtACR1. However, this finding is not clearly reflected in the mechanistic models presented in this work. This point should be addressed, and if necessary, the mechanistic models and Figures 2 and 6 should be revised accordingly.

It is correct that recent UV/VIS measurements report an involvement of an O-intermediate in the conducting state.

However, the publications (Refs. 50 and 51) have a quite unusual definition of the O-intermediate. Since the absorption maximum of the intermediates is the only observable in the visible, they use the absorption maximum to label the intermediates. Thus, all intermediates that absorb at long-wavelength around 610 nm are referred to as "O-intermediates". In contrast, we use the common definition based on the bR photocycle with the O intermediate as the last intermediate between N and ground state. It follows the non-conducting M and N intermediates in the course of the photocycle. Thus "our" O-intermediate is clearly a non-conduction intermediate.

Page 6, Lines 218-221:

"This interpretation is further supported by the increased formation of the band at 1691 cm^{-1} during stronger light exposure (see Figure 4f/g), which reflects alterations in the secondary structure during channel opening and clearly supports the buildup of the postulated pre-open state of the ion channel."

If 30 seconds of illumination enhances the 1691 cm^{-1} peak representing the channel opening, one might expect that longer exposure would further open the channel and increase the photocurrent. However, the stationary current is weaker than the peak current, as shown in Figure 1b. Please clarify this apparent inconsistency.

We believe we have two misunderstandings here. First, the spectra in Fig.4 are amplitude spectra, meaning they reflect changes during a specific reaction step, in this case the transition from the N/O intermediate to the ground state, both of which are non-conductive. The increase of the absorption at 1691 cm^{-1} , assigned to conformational changes during channel opening, is indicating the presence of said conformational changes.

Second, our definition of "pre-open" state was not clear enough. According to reviewer 3 we could also call it "pre-closed" state. We define it as a state that does not conduct ions, but there are structural features that are in between the open state and the closed state.

An increase in the photocurrent would be highly unlikely regardless of the repeated excitation, as all channels are fully open when the channels are first excited and cannot be opened any further. However, since only the O-intermediate can be excited due to the retinal configuration, a mixture of conductive and non-conductive states prevails in the steady state due to the life-time of the non-conductive states and back reactions in the photocycle, which lead to a slight decrease in conductivity.

We have reworded the corresponding paragraph to make it clearer that the changes described are the transition from the N/O intermediate to the ground state and have also redefined the 'pre-open' state to avoid misunderstandings:

'The decrease in the E68 band at 1708 cm^{-1} (see Figure 4 f/g) during the transition from the N/O to the ground state demonstrates a lower degree of reprotonation of E68 in the N/O intermediate with increased light exposure. Since the fast deprotonation of E68 is associated with the channel opening process^{66,85}, the deprotonated state of glutamate in the N/O intermediate may facilitate opening more readily than the state reached after full deprotonation and reprotonation, as observed after a single flash activation. We assume that the central constriction site may therefore exist in a "pre-open state", which allows for rapid channel opening upon excitation from the O intermediate. Although the pre-open state probably does not conduct ions, there are structural features that are in between the open state and the closed state, enabling the proteins to return faster to the conducting state.'

This interpretation is further supported by the increased formation of the band at 1691 cm^{-1} during stronger light exposure (see Figure 4 f/g). A higher absorbance at 1691 cm^{-1} indicates an increased presence of the assigned group, in this case alterations in the secondary structure which only occur during channel opening, thereby supporting the buildup of the postulated pre-open state of the ion channel.'

Minor Points

Page 4, Lines 132-133:

"Channel opening itself and the conducting state, visible at 1691 cm^{-1} , follow the same pattern as described in literature for the wild type."

The strong similarity of the 1691 cm^{-1} peaks, which represent the channel-open state, between GtACR1 WT and Q46E seems inconsistent with the large difference in peak current observed in Figure 1b. Please

clarify why the Q46E mutant, which exhibits a similar channel-opened structure to the WT, has a much smaller peak photocurrent.

One of our key findings is the high similarity of the amplitude spectra of WT and Q46E, showing that, against the initial assumption, they have the same mechanism. It is correct that the high similarity alone cannot explain the very different conductance. However, the analysis of the lifetimes of the open and the closed intermediates in combination with the possibility to excite the O intermediate makes an explanation possible. While the closed M-intermediate in the WT has only a short lifetime, this lifetime is about 10 times longer in the mutant. Thus, in steady state conditions during continuous illumination, a large fraction of the channel is in a closed state that cannot be excited for the mutant.

We discussed this in more detail on page 5, Line 145-160:

'The extensive similarity between the UV/Vis and FTIR spectra suggests that GtACR1 WT and Q46E exhibit similar reaction mechanisms, except for the significantly decelerated photocyclic reaction rates in the variant. Since a syn-cycle is ruled out, and the conductive state of the channel is relatively short (even the slow channel-closing lifetime is about 965 ms), compared to the long return time to the excitable ground state, due to the increased T_4 - T_6 half-lives, we conclude that the electrophysiologically observed attenuation is due to the accumulation of intermediates preceding the photoactivatable ground state.'

'Given the above explanation, it is questionable why attenuation is almost absent in GtACR1 WT upon continuous illumination (see Figure 3). The channel closes completely with a half-life of only 107 ms, compared to the long half-life of $T_6 = 4.4$ s for the return to the ground state. This should immediately enrich the non-conducting N/O intermediate, leading inevitably to conductive attenuation. However, as no attenuation occurs in WT GtACR1, we hypothesize that continuous photocycling is driven by the N/O intermediate, specifically through photoactivation of the O intermediate, whose retinal cofactor is already in the all-trans configuration. This is a precondition that is otherwise only satisfied by the dark ground state.'

Page 7, Lines 239-240:

"...as the positive bands exhibit only minor changes."

In Figure 5, many significant changes are observed in the positive bands. Is this a typographical error, and should it be "negative bands"?

In this context, we are actually referring to the positive bands. The shifts all amount to 5-10 cm^{-1} , which is considered minor when talking about FTIR amplitude spectra, especially in the regions where these shifts were observed. We couldn't detect a change significantly enough to indicate a second photocycle, which is why this formulation has been chosen.

We understand how this can be confusing and have rephrased the sentence for more clarity and to avoid confusion:

'Although shifts of $4 \text{ cm}^{-1} - 8 \text{ cm}^{-1}$ occur, these are comparatively small for IR shifts, which is why it can be assumed that GtACR1 enters a related photocycle upon excitation of the O intermediate.'

Page 7, Lines 251-252:

"... the pore is not fully closed and therefore does not need to open at the beginning of the photocycle."

If the pore is not fully closed, a leak current, whose sign should change depending on the holding potential during electrophysiological measurements, would be expected even before the photocycle begins. However, to the best of my knowledge, such a leak current has not been observed for GtACR1, except for the E68R mutant (Sineshchekov et al., PNAS (2015), 112, 14236-14241). Please clarify this point.

As written above our definition of the pre-open state describes the state of the pore upon excitation from the O-intermediate. Before the initial excitation, the pore is completely closed and opens with the first excitation; accordingly, no leakage will be detected. In the case of excitation from the O-intermediate, the pore does not close completely due to the shortened photocycle, so that the opening from the shortened photocycle can take place more quickly because it is still partially open.

We have reformulated the sentence and hope that this makes the statement easier to understand:

'The decrease in this band under strong illumination, where efficient ion current is observed, suggests that the pore is not fully closed upon excitation from the O intermediate and therefore does not need to open at the beginning of the shortened photocycle.'

Page 8, Lines 264-265:

"... the L2 and M intermediates. During these states, the pore is in a non-conductive state, and..."
Does L2 not represent a conductive state, as shown in Figure 2?

We understand that this sentence is misleading and have therefore rephrased:

'The findings also apply to the GtACR1 variant Q46E. The high attenuation of the photocurrent observed in this variant, compared to the WT, can be attributed to the prolonged presence of the L₂ and M intermediates. With the decay of the L₂ intermediate and the formation of the M intermediate, the channel enters a non-conductive state and since the retinal remains in the 13-cis configuration, no new excitation can occur.'

Page 8, Lines 295-296:

"... new approaches for improving CrChR2 for optogenetic applications, aiming to increase the channel's conductivity, can be found based on these findings."

It is unclear what specific approaches could be derived from the findings of this study to improve CrChR2 for optogenetic applications. Please provide a more concrete explanation.

We attempted to provide a more concrete explanation which specific approaches could be derived from our findings:

'Furthermore, the hypothesis that the O intermediate, retaining an all-trans retinal configuration and a "pre-open" state of the central gate, might allow for rapid channel reopening could provide a new framework for understanding sustained ion conduction under continuous illumination. A thorough understanding of the gating mechanisms and attenuation in light-activated ion channels is pivotal for the rational design of next-generation optogenetic tools.'

Page 10, Lines 351-352:

"In addition, 10 µl all-trans retinal in 10 ml 100 % methanol is added to each flask."

The meaning of "10 μ l all-trans retinal in 10 ml 100% methanol" is unclear. Does this mean that 10 μ l of a stock all-trans retinal solution was added to 10 ml of methanol? Please rephrase this sentence for clarity.

We have rephrased the sentence and hope the meaning is clearer now:

'Furthermore 10 μ l of a stock solution all-trans retinal was added to 10 ml 100 % methanol and the mixture is added to each flask'

Page 11, Lines 368 and 384:

"maltosid" should be corrected to "maltoside."

We have changed 'maltosid' to 'maltoside'

Figure 1b:

The photocurrent intensity of ChRs can vary depending on cell type, ion composition, membrane potential, light conditions, etc. Were the photocurrents shown in Figure 1b recorded under identical experimental conditions? Otherwise, comparing their absolute intensities may not be meaningful. If all measurements were conducted under the same conditions, please specify the experimental details.

We understand that the direct comparison of the photocurrents, as shown in Figure 1b, may not appear meaningful at first glance, as the measurements were carried out by two different groups, which is why identical conditions are not guaranteed.

However, we compare the *CrChR2* WT and the *GtACR1* variant Q46E in relation to the *GtACR1* WT. In the publication by Govorunova et al. the *CrChR2* WT and *GtACR1* WT are compared whereas Kim et al. compared the *GtACR1* WT and *ACR1* variant Q46E. Identical measurement conditions were used in the papers themselves, so that a comparison of the *CrChR2* WT and the *GtACR1* variant Q46E is possible after normalization of the *GtACR1* WT. Our statement, that the *GtACR1* variant Q46E shows a similar attenuation profile as the *CrChR2* WT is therefore independent of the measurement conditions.

To make this clearer, we have added it to the figure legend:

'Photocurrents of CrChR2 WT, GtACR1 WT and GtACR1 Q46E, normalized to the GtACR1 WT. The shown photocurrents are a response of CrChR2 and GtACR1 (data replotted from Govorunova et al., 2015, Fig. 1D12) and GtACR1 Q46E (data replotted from Kim et al., 2018, Extended Data Fig. 8) to a 1 s light pulse.'

Figures 3c and 4a, and Supplementary Figure 2:

Labeling the prominent peaks would enhance readability.

We have labelled the peaks discussed in the kinetics analysis to improve readability.

Figure 3:

Figure 4:

SI Figure 2:

Figure 5:

How were these excitation spectra obtained? Please describe the experimental details used to acquire these spectra.

We have included a section in the methods describing the experimental details that were used to obtain the excitation spectra:

'The excitation spectra $a_0(\nu)$ describe the absorbance differences obtained by comparing the absorbance before the laser flash with the absorbance extrapolated to $t=0$ immediately after the laser flash, before the resolved transitions take place.'

Figure 6:

Indicating the protonation state of E68 in each intermediate would help clarify the proposed mechanism.

We have added the protonation state of E68 in each intermediate to Figure 6 for clarity

Reviewer #2 (Remarks to the Author):

The authors present the discovery of a novel secondary-photon-activation photochemical mechanism that greatly increases the conductance by the light-gated anion channelrhodopsin GtACR1. Their results are compelling, and the second-photon photochemical reactions are revealed by their innovative experiments and their expertise with time-resolved kinetic FTIR, and their comparison of GtACR1 and the cation channel CrChR2 structures and photochemical reactions, two widely used channelrhodopsins as optogenetic tools. The enhancement of ion currents through the membrane by the mechanism they unraveled explains the much greater conductance of GtACR1 in continuous light compared to that of CrChR2 in the same condition.

The results presented are of immediate relevance to researchers applying optogenetics (e.g. in neuroscience or cardiac cell research, which involve channelrhodopsins). The discovery of this new mechanism also establishes a new concept to be considered by investigators of any photoactivated protein that undergoes cyclic photochemical reactions (photocycles), therefore the impact of this paper reaches also a broad audience in photochemistry and photobiology. In continuous light, secondary photochemistry occurs in GtACR1, which entails absorption of a second photon by a late photocycle intermediate that drives the protein back to an early photocycle intermediate conformation. This early intermediate quickly converts to the conformation that opens the channel. Therefore, the authors have discovered an elegant, novel mechanism produced by evolution that avoids disrupting, but rather enhances, the effective single-photon photocycle by creating a second-photon photocycle within the single-photon photocycle. The result is that the photochemical reactions produce a longer lasting conducting open-channel conformation in the protein.

The writing is concise and well written. The research is clearly and accurately described both in the history of channelrhodopsins and the new exciting discovery.

We thank the reviewer for the kind feedback on our manuscript.

Reviewer #3 (Remarks to the Author):

The manuscript under review addresses an important issue in optogenetics—the phenomenon of photocurrent attenuation in anion-conducting channelrhodopsins—and seeks to elucidate its molecular basis by focusing on GtACR1 and a Q46E mutant variant. The authors present a comprehensive FTIR spectroscopic investigation that attempts to correlate subtle changes in retinal configuration and protein conformational dynamics with the functional phenomenon of attenuation. This is a timely subject, given that a thorough understanding of the gating mechanisms and attenuation in light-activated ion channels is pivotal for the rational design of next-generation optogenetic tools.

The work is conceptually motivated by the challenge of overcoming the attenuation observed in cation channelrhodopsins such as CrChR2. By comparing GtACR1 with its mutant Q46E, the authors draw attention to the role of the central gate—a region that is conserved between ChR2 and ACR1—in modulating ion conductance and the kinetics of channel closing. The manuscript carefully documents the alterations in photocycle reaction rates in the mutant relative to the wild type, and it proposes that the accumulation of intermediates preceding the photoactivatable ground state is responsible for the observed attenuation. The hypothesis that the O intermediate, retaining an all-trans retinal configuration and a "pre-open" state of the central gate, might allow for rapid channel reopening is intriguing and, if substantiated, could provide a new framework for understanding sustained ion conduction under continuous illumination.

Despite the conceptual clarity and logical progression in the authors' argument, several critical issues remain that require further attention, therefore I would recommend major revision of the manuscript. The experimental design relies exclusively on FTIR spectroscopy to probe the dynamics of the photocycle and does not include complementary electrophysiological measurements. In previous studies on CrChR2, a combination of FTIR, electrophysiology, and Raman spectroscopy was employed to correlate spectral changes with functional currents (Kuhne et al PNAS, 2019, doi.org/10.1073/pnas.1818707116). The absence of such direct functional data in the current study limits the ability to unambiguously link the spectroscopic signatures to channel conductance, particularly regarding the proposed rapid re-excitation of the O intermediate. Direct evidence, such as the dependence of photocurrent magnitude on illumination intensity or pH, would greatly strengthen the authors' claims regarding the mechanistic origin of attenuation.

Like Reviewer 1, Reviewer 3 also has raised concerns as we didn't include complementary electrophysiological measurements in our manuscript. We agree that electrophysiological measurements under similar light conditions would enhance the quality of the manuscript.

As mentioned above, we do not have the necessary equipment in our laboratory to carry out these measurements ourselves, and replicating similar lighting conditions in another laboratory is very complicated due to the complexity of the laser we use.

However, electrophysiological measurements are available for all measured protein variants, which are in full agreement with our spectroscopic data. Our results explain all features of the electrophysiological experiments, including the extent of the attenuation and the peak at the beginning, very well.

If the L_2 intermediate is the only conducting intermediate, and the return to the excitable ground state does take 4.4 s, as postulated by Dreier et al., immediate current attenuation should take place. As this clearly isn't the case, there must be a cause for the high effectiveness of the channel, such as another photoactive intermediate. We identified the O-intermediate as this photoactive intermediate. As only the O-intermediate can be excited due to the retinal configuration, a mixture of conductive and non-conductive states prevails in the steady state due to back reactions in the photocycle, which lead to a slight decrease in conductivity.

In order to verify that the processes observed under the single/multiple flash setup also take place under continuous illumination, we also carried out measurements with illumination conditions similar to electrophysiology, which confirmed our assumptions.

In addition, our previous article published in *Communications Biology* (Dreier et al., <https://doi.org/10.1038/s42003-021-02101-5>) was also entirely based on FTIR measurements and only referred to previously published electrophysiological data. The publication of our article could motivate electrophysiological work on the re-excitable O-intermediate from the dependence of photocurrent magnitudes on light conditions and pH, which would further stimulate scientific exchange.

The choice of mutation in the central gate also raises some concerns. The substitution of a glutamine with a glutamate in the Q46E mutant is interesting, yet the rationale behind this specific change is not fully justified, especially in light of the observation that the mutant still exhibits a similar attenuation phenomenon.

The *GtACR1* variant Q46E was selected because of the great similarity of the photocurrent to that of the *CrChR2* WT (see Figure 1b):

Current attenuation of *CrChR2* is a major problem in optogenetics, as high photocurrents and low current attenuation are highly desired, as observed in *GtACR1* WT. Kuhne et al. has identified the low-conducting *syn*-cycle as the cause of current attenuation in *CrChR2*. Based on the high sequence identity of both proteins in the central gate, which is described on page 4, lines 104-115, a similar reaction mechanism can be assumed. As mentioned in the original manuscript (page 4, lines 116-122), the high similarity of the photocurrent of the *GtACR1* variant Q46E with the *CrChR2* WT is the reason why we selected this variant. Our initial assumption was that the high similarity was due to the presence of a *syn*-cycle in variant Q46E that we could have identified using FTIR spectroscopy, which would have made the residue a promising target to compensate for the poorly conducting *syn*-cycle in *CrChR2*. Unfortunately, we were unable to detect a *syn*-cycle, which prompted us to investigate further into the reason for the observed attenuation.

One would expect that replacing a neutral residue with a charged one could significantly alter local electrostatics, and therefore, a clear dependence on pH should be evident. However, the manuscript does not specify the pH conditions under which the FTIR experiments were performed, leaving an important experimental variable unaddressed.

We clarified the pH conditions within both the main text and the corresponding method section:

‘To test this hypothesis, rapid-scan FTIR measurements were carried out at ambient temperature, with a neutral pH (7.5) and compared to wild type measurements’

‘The resulting suspension containing proteoliposomes and buffer is further processed by separating the proteoliposome suspension from the beads and centrifuging for 10 minutes at 20000 rpm. The resulting pellet is then resuspended in measuring buffer (20 mM HEPES, 100 mM NaCl, pH 7.5) and ultracentrifuged at 55000 rpm for 3 h using a Thermo Scientific MTX150 micro-ultracentrifuge with a S55-A2 rotor.’

A more detailed discussion of the pH sensitivity of the system is warranted, as the protonation states of residues such as E68 are central to the proposed model of the "pre-open" state.

We agree with the reviewer that a detailed discussion of pH sensitivity in the context of the protonation state of E68 is very useful. Therefore, we took the reviewer's feedback as an opportunity to perform

measurements of the GtACR1 WT at different pH values. However, we encountered the problem that the sample proves to be unstable at low (5.5) and high (8.5) pH values. The corresponding measurements show significantly reduced amplitudes for all three reaction rates, indicating a high percentage of denatured protein:

pH 5.5:

pH 8.5:

Unfortunately, due to the high proportion of denatured protein, no reliable statements can be made about the observed changes.

Minor comments:

Line 15

"Optogenetics is a method to regulate cells using light."

Possibly, it might be better to mention not only cells, but also tissues and organisms.

We added tissues and organisms to the abstract.

Line 16

It is appropriate bet to mention not only neurons but also other targets of optogenetics

Unfortunately, the abstract is limited to 150 words, therefore we are not able to go into all of the details of optogenetics in the abstract. We go into more detail in the introduction (Page 2, Line 34-40):

'For example, ectopically expressed CrChR2 can control action potential firing with high temporal and spatial resolution in mammalian neurons^{34,35}. In addition to neurons, a variety of other cell types such as muscle cells^{36,37}, immune cells^{38,39}, endocrine cells^{36,40} or stem cells^{41,42} can also be controlled by optogenetic tools. Recent advances highlight the vast potential of optogenetics for various medical applications, including the treatment of blindness^{31,43}, Parkinson's disease^{44,45}, cardiac arrhythmia^{46,47} and the regulation of insulin secretion^{36,40}.'

Line 17

Might I suggest changing "very low" to "lower"?

We exchanged 'very low' to 'lower'

Line 18

Could you explain what do you mean a "branched photocycle"? Could it be two different photocycles of different fractions of rhodopsins (like in ChR2 (Kuhne et al PNAS, 2019, doi.org/10.1073/pnas.1818707116))?

The phrase „branched photocycle” is directly referring to the photocycle model published by Kuhne et al. as they use this phrase to describe their photocycle model with two different photocycles:

'We resolve this by proposing a branched photocycle explaining electrical and photochemical channel properties and establishing the structure of intermediates during channel turnover.'

Again, due to the word limitation in the abstract it is not possible to go into further detail here, but we described the photocycle model in detail in the introduction (see Page 2, lines 59-71):

'In the case of CrChR2, blue light excitation ($\lambda = 470$ nm) induces two parallel photoreaction cycles (see Supplementary Figure 1)⁷³. One of these is the so called "dark-adapted" anti-cycle, characterized by the exclusive occurrence of a C=N-anti configuration of retinal and a well-conducting open state that decays relatively quickly. The other one is the slowly decaying "light-adapted" syn-cycle, characterised by the 13-cis, C=N-syn configuration of retinal and the presence of poorly conducting photoproducts⁷³. Electrophysiological measurements show that under continuous illumination, the highly conductive anti-cycle is associated with a current peak that drops rapidly - referred to as attenuation - when CrChR2 switches to the low-conductivity syn-cycle^{66,73} (see Figure 1 b). Both cycles are spectroscopically distinguishable due to their cycle-specific retinal configurations. The spectroscopic marker band of the anti-cycle is found at 1188 cm^{-1} representing the C=N-anti retinal configuration, while the marker band for the syn-cycle is at 1154 cm^{-1} , corresponding to the C=N-syn retinal configuration⁷³.'

It would be rational to mention in the Introduction other examples of photoswitchable photocycle intermediates for other microbial rhodopsins, for example for bacteriorhodopsin.

We have added other examples of photoswitchable photocycle intermediates for other microbial rhodopsins to the introduction:

Microbial rhodopsins are integral membrane proteins with seven transmembrane helices and the cofactor retinal. Due to the photoactivation of the retinal, they can fulfill several functions, including proton pumps, ion pumps and ion channels. The exact atomic mechanisms within the photocycles have been of scientific interest for decades. Starting with the "hydrogen atom of biophysics" bacteriorhodopsin¹⁻¹¹, the research has expanded to all kinds of microbial rhodopsins, such as halorhodopsin¹²⁻¹⁴, schizorhodopsins¹⁵⁻¹⁹ and channelrhodopsins²⁰⁻²³.

Line 124

During describing the procedure of global fit analysis it would be correct to mention original works by Chizov et. al about this method (Chizhov, I. et al., Biophysical Journal, Volume 71, Issue 5, 2329 – 2345)

The citations have been added to both the main text and the corresponding method section:

'The major difference between the WT and the variant lies in the reaction rates, expressed in the half-lives derived from global fit analysis⁸¹⁻⁸⁴, as the GtACR1 variant Q46E is significantly slower than the wild type'

'For the analysis of the time-resolved data a global fit⁸⁴ in MatLab (The MathWorks, MA, USA) was used⁸⁴, as described by Chizow et al. ⁸¹⁻⁸³.'

Line 220

In the sentence "alterations in the secondary structure during channel opening and clearly supports the buildup of the postulated pre-open state of the ion channel." Instead of using the word "clear", please, explain why it "supports"?

We understand and agree that the statement was not sufficiently formulated to support the thesis. We have rewritten the sentence to explain in more detail how the alterations in the secondary structure during channel opening support the pre-open state:

'This interpretation is further supported by the increased formation of the band at 1691 cm⁻¹ during stronger light exposure (see Figure 4 f/g). A higher absorbance at 1691 cm⁻¹ indicates an increased presence of the assigned group, in this case alterations in the secondary structure which only occur during channel opening, thereby supporting the buildup of the postulated pre-open state of the ion channel. The environmental changes may also affect the neighboring protonated Schiff base, which explains the observed shift of the band of protonated 13-cis C=N-anti retinal at 1184 cm⁻¹ (see Figure 4 d/e).'

Line 278

Why this intermediate is called "pre-opened" state? For example, for CrChR2 it was shown that opening of the channel for protons occurs much earlier in the photocycle (during M intermediate), and O intermediate is the last one in the photocycle. Maybe this intermediate could be called "pre-closed" state?

Our definition of the pre-open state describes the state of the pore upon excitation from the O-intermediate. Before the initial excitation, the pore is completely closed and opens with the first excitation. In the case of excitation from the O-intermediate, the pore does not close completely due to the shortened photocycle, so that the opening from the shortened photocycle can take place more quickly because it is still partially open, although no current can be detected. Technically speaking, it could also

be referred to as a ‘pre-closed’ state, as the channel would be closed if no further excitation were to take place. However, we are referring to the state of the channel in the context of repeated or continuous excitation. In this case, the state of the channel would precede the open state, which is why we have chosen to refer to it as a ‘pre-open’ state.

In Fig.6 you represent a photocycle model with K-intermediates, but as I understood, experimental data in this work was obtained starting from the L-intermediate (due to time resolution limits). Could you, please, mark also in the Fig.6 that a short-lived K-like intermediate is hypothetical (not directly measured experimentally).

We have changed the box around the intermediate K' from a solid box to a dotted one, to symbolize, that the intermediate is just hypothetical. Furthermore we added a statement to the figure legend, that K' is not measured experimentally, but assumed:

‘We hypothesize, that with the transition to the 13-cis configuration a short-lived K-like intermediate, which we refer to as K', is reached before entering the known photocycle at the L₁/L₁' intermediate, followed by channel opening. As this intermediate is not experimentally measured, it is represented by a dotted box.’

Reviewer #1 (Remarks to the Author):

Although the authors have revised many parts of the manuscript in response to my comments, unfortunately, two major concerns remain insufficiently addressed.

Regarding the request from both Reviewer 3 and myself for electrophysiological experiments to support the conclusions drawn from FTIR data, the authors explained the difficulties in performing such measurements and instead added a detailed explanation of the photocurrent data in Figure 1, referencing the original studies. However, while the photocurrents in the study by Govorunova et al. were measured at a holding potential of -60 mV in HEK293FT cells 10 to 19 days after transfection, Kim et al. performed patch-clamp recordings in HEK293 cells 24–48 hours after transfection. Therefore, the protein environments differ considerably, making normalization using the GtACR1 wildtype data difficult. In fact, in the latter case, the negative photocurrent shows strong attenuation, which was not observed under the experimental conditions used in the former study. Thus, reproducing the data from these studies in Figure 1 is not appropriate and misleading for the readers. The photocurrents of the three proteins should be recorded under the same experimental conditions, and an electrophysiological study supporting the authors' conclusions is necessary. If this is difficult using the authors' current system, they could consider collaborating with another group equipped with the appropriate instrumentation for the purposes of this study. Although the authors state that their light illumination pattern is difficult to reproduce under electrophysiological conditions, many electrophysiology laboratories are capable of using nanosecond 532-nm laser pulses or 525-nm LEDs as stimulation light sources. Therefore, this does not justify avoiding such collaboration. Otherwise, since the relationship between the FTIR-detected structural changes and the channel-opening mechanism cannot be fully understood without electrophysiological data, I recommend submitting the manuscript to a more specialized journal.

We agree with the concerns of reviewer 1 that Figure 1 is misleading regarding the comparability of the electrophysiological data.

Therefore, we have adapted Figure 1 to show the comparison of the CrChR2 WT and the GtACR1 WT, both measured by Govorunova et al. at a holding potential of -60 mV in HEK293FT cells 10 to 19 days after transfection (Figure 1b), and the comparison of the GtACR1 WT and the variant Q46E measured by Kim et al. with patch-clamp recordings in HEK293 cells 24–48 hours after transfection (Figure 1c). We hope that the expanded data will be considered adequate and no longer misleading.

Figure 1: Structural and functional comparison of CrChR2 and GtACR1 a) Structural alignment of the central constriction site residues of CrChR2 (green amino acid carbon atoms) from PDB-ID 6EID21 and GtACR1 (grey amino acid carbon atoms) from PDB-ID 6EDQ22. The retinal (black carbon atoms) is taken from GtACR1. Nitrogen atoms are coloured blue and oxygen atoms red. b) Photocurrents of CrChR2 WT and GtACR1 WT. The shown photocurrents are a response of CrChR2 and GtACR1 (data replotted from Govorunova et al., 2015, Fig.

ID12) to a 1 s light pulse at a holding potential of -60 mV in HEK293FT cells 10 to 19 days after transfection. CrChR2 shows the characteristic decay of the photocurrent upon continuous illumination. GtACR1 produces a significantly larger photocurrent compared to CrChR2 and does not display the described inactivation of the photocurrent. However, channel closing is much slower in GtACR1 compared to CrChR2. c) Photocurrents of GtACR1 WT and GtACR1 Q46E (data replotted from Kim et al., 2018, Extended Data Fig. 8). The shown photocurrents are a response to a 1.5 s light pulse in patch-clamp recordings of HEK293 cells 24–48 hours after transfection. The GtACR1 WT shows a significantly larger photocurrent compared to the GtACR1 variant Q46E. Furthermore, GtACR1 Q46E exhibits strong current attenuation.'

However, our initial statement, that both the CrChR2 WT and the GtACR1 variant Q46E exhibit a lower photocurrent compared to the GtACR1 and show an inhibition of the photocurrent that is not observed in the GtACR1 WT, remains valid as they are recorded under the same conditions. The comparison between GtACR1 and CrChR2 electrophysiological data is in our manuscript used as motivation for the impact of the work within the introduction. Therefore, we believe a qualitative comparison is sufficient and for the purpose of this manuscript no rerecording of electrophysiological data is necessary. Further, all protein variants discussed in our manuscript are well described electrophysiologically in the literature. Our spectroscopic data is in full agreement with these electrophysiological measurements as discussed in detail in the manuscript. To further improve the comparability between our FTIR data and the existing electrophysiology data in the literature we included FTIR measurements with illumination conditions similar to the electrophysiological measurement, to show that the processes are the same. The results after 30 seconds of continuous illumination are identical to the ones after multiple illuminations and nicely explain all the features of the electrophysiological experiments, including the amount of attenuation and the peak in the beginning. Reviewer 3 agreed with this strategy.

Therefore, the results presented in the manuscript stand for themselves. We are convinced that further electrophysiological experiments go beyond the scope of this manuscript.

In Answers to the Reviewers: "...we performed lower intensity experiments at the beginning of the studies to ensure that we are measuring in saturation. We measured the samples at half the laser intensity used in the data shown, resulting in comparable excitation amplitudes indicating that the yield of excited proteins is indeed in saturation... Furthermore, reviewer 1 is correct that intensity and illumination time are crucial factors of how many proteins are converted into photointermediates. However, the referred paper by Malakar et al. used a very short laser pulse with 100 fs. In our experiments, we used a 12 ns pulse (120-fold longer). The lifetime of the retinal excited state is about 500 fs. Thus, proteins that are return to the ground state without isomerization can be excited again to enter the photocycle".

This point seems different from the one I raised. My concern is the suppression of photoproduct accumulation due to back-isomerization from the K intermediate (13-cis) to the initial all-trans state, not from the retinal excited state. This back-isomerization also occurs when a nanosecond laser pulse is used as the excitation source. Signal saturation does not guarantee 100% accumulation of the photoproduct. During strong laser pulse illumination, a photo-equilibrium between the initial and K-intermediate states is established. The ratio between these states is governed by the quantum yields of all-trans to 13-cis isomerization (from the initial state) and 13-cis to all-trans isomerization (from the K intermediate), the latter being higher than the former as reported in Malakar et al., J. Phys. Chem. Lett. (2022) 13, 8134–8140, as well as by their respective extinction coefficients at the excitation wavelength. Please take these considerations into account, and I recommend the authors carefully evaluate the populations of photointermediates.

We thank the reviewer for rising this point. Indeed, we did not consider a possible backreaction from K to the ground state. As the reviewer correctly pointed out, this back-reaction is not covered by our saturation experiments. Therefore, we spectroscopically evaluated the populations of the

photointermediates. For this we determined the overall protein amount in the sample by means of the amide II absorption. Next, we determined the amount of protein undergoing the photocycle by means of the difference absorption of the retinal. The calculation is shown in detail below and added to the Supplement of our manuscript. Contrary to our initial assumption, only about 40 % of the sample enters the photocycle after one flash. Thus our data in Figure 4 shows a mixture of excitation from the O-intermediate and the ground state and the spectral changes comparing excitation from the ground state or the O intermediate are about twice as much as seen there. However, this new finding better explains the stronger changes after the 5 flash experiment compared to the two flash experiment as in each flash 40% of the remaining ground state protein enters the photocycle. After 4 flashes (and thus before the 5th flash) the remaining ground state protein fraction is $0.6^4 = 0.13$.

We have adapted the main text and referred specifically to the mixture. The statement that the O-intermediate is photoactive remains valid despite the low level of excitation.

'Under our measurement conditions, all molecules are excited, as demonstrated by experiments at lower laser intensities, resulting in almost the same absorbance changes. Thus, we are working under saturation conditions. However, it must be taken into consideration that back-isomerization from the K intermediate (13-cis) to the initial all-trans state is possible¹ and a photo-equilibrium between these two states is established. We determined the position of the equilibrium by comparing the overall protein amount in the sample by means of the amide II absorption and the amount of protein undergoing the photocycle by means of the difference absorption of the retinal. This shows that about 41% of the sample enters the photocycle (see Supplemental Note 1). Upon a second excitation, the spectra would therefore contain a mixture of proteins excited from the O intermediate state and the ground state.

To test whether the hypothesis of additional excitability of the O intermediate applies to the GtACR1 WT, the proper timing for repeated excitation must be adjusted to ensure that the O intermediate is precisely targeted. Based on the known photocycle time constants, it was calculated that 1 second after the initial activation from the ground state, the M intermediate (which decays with a slow channel-closing half-life T_5 of 107 ms) is reduced to less than 1 % of its initial value (see Supplementary Note 2), indicating that over 99 % of the protein has reached the N/O intermediate. Similarly, it was calculated that after 1 second of activation 96 % of the proteins within the photocycle are still remaining in the N/O intermediate (see Supplementary Note 3).

At this time point, a second flash was applied to initiate another photocycle, which was monitored by time-resolved FTIR spectroscopy.

Under consideration that 41% of the proteins enter the photocycle, after 1 second, about 38% are in the N/O intermediate. Global fit analysis of...'

'As calculated above, after 1 s about 2/3 of the sample is in the ground state and 1/3 in the O intermediate. Therefore, the spectra contain a mixture of proteins that were excited from the O-intermediate and the ground state. To increase the amount in the O-intermediate, spectral development was monitored under different illumination regimes: after one flash, two flashes, and five flashes (with 1 Hz repetition rate; see Figure 4 d/e), as well as after 30 s of continuous illumination (see Figure 4 f/g). With the 5th laser flash about 87 % of the sample is excited from the O intermediate (Supplementary Note 1).'

Further the following was added to the supplementary information:

In order to be able to make reliable statements about the excitability of the O intermediate, it was necessary to determine how much of the sample is excited upon one flash.

With help of the background spectrum, which reflects the entirety of the proteins in the sample in the amide II band, and the excitation spectrum, which shows the excited proteins, the percentage excitation could be calculated using the Lambert-Beer law as follows:

In a first step a solution containing 100 mM all-*trans* retinal was measured on a micro ATR and the molar extinction coefficient of retinal was calculated approximately using the Lambert-Beer law

$$\epsilon_{Ret} = \frac{A_{1375} \text{ cm}^{-1}}{c \times d} = \frac{0.213}{0.1 \text{ M} \times 0.0009527 \text{ cm}} = 2200 \frac{1}{\text{M} \times \text{cm}}$$

A similar value is found in the literature².

Whereby the effective penetration depth (d_e) of the micro-ATR cell was calculated with the penetration depth of one reflection (d_p) and the number of total reflections (N):

$$d_p = \frac{\lambda}{2\pi n_1 \sqrt{\sin^2 \theta - \left(\frac{n_2}{n_1}\right)^2}} = \frac{7256 \text{ nm}}{2\pi \times 2.4 \sqrt{\sin^2(45^\circ) - \left(\frac{1.3}{2.4}\right)^2}} = 1058 \text{ nm}$$

$$d_e = N \times d_p = 9 \times 1058 \text{ nm} = 9527 \text{ nm}$$

The extinction coefficient of the amide II band (ϵ_{AII}) is calculated from the number of peptide bonds (541) of the protein and the extinction coefficient of a single molecular vibration

$$\epsilon_{AII} = 350 \frac{1}{\text{M} \times \text{cm}} \times 541 = 189350 \frac{1}{\text{M} \times \text{cm}}$$

The amide II absorbance of the total protein was determined with

$$A = -\lg \frac{I}{I_0} = -\lg \frac{0.32}{0.66} = 0.31$$

The absorbance change of the excited protein could be read in the excitation spectrum at 1235 cm⁻¹:

$$\Delta A = 0.0017$$

Finally, the ratio of excited protein to total protein can be calculated:

$$ratio\ excitation = \frac{c_{excited} \times d}{c_{total} \times d} = \frac{\Delta A}{A} \times \frac{\epsilon_{A2}}{\epsilon_{Ret}} = \frac{0.0017}{0.31} \times \frac{189350}{2200} = 47\%$$

In this way, the excitation of 6 samples was calculated, whereby an average excitation of the sample upon one flash of about 41 % +/- 5 % was determined.

Subsequently, before the 5th flash $0.6^4 = 13\%$ of the sample remain in the ground state, which means that 87 % are excited from the O intermediate.

Reviewer #3 (Remarks to the Author):

I carefully read the revised manuscript and the answers of the authors, I am quite satisfied with their revision and I think that this manuscript can be useful for scientific community. I recommend it for publication.

We thank the reviewer for the kind feedback on our manuscript and his assistance to improve it.

References

- 1 Malakar P, Das I, Bhattacharya S, et al. Bidirectional Photochemistry of Antarctic Microbial Rhodopsin: Emerging Trend of Ballistic Photoisomerization from the 13-cis Resting State. *J Phys Chem Lett.* 2022;13(34):8134-8140. doi:10.1021/acs.jpcllett.2c01974.
- 2 Andresen ER, Hamm P. Site-specific difference 2D-IR spectroscopy of bacteriorhodopsin. *J Phys Chem B.* 2009;113(18):6520-6527. doi:10.1021/jp810397u.